

# Hybridisation of brownboost classifier and glowworm swarm based optimal sensor placement for water leakage detection

Rejeesh Rayaroth[1], Sivaradje Gopalakrishnan[1],

[1]Department of Electronics & communication Engineering, Pondicherry Engineering College, Puducherry, 605014, India

*Correspondence to*: Rejeesh Rayaroth (rejeeshrayaroth@pec.edu)

**Abstract.** Water Distribution System distributes the water to customer with the better quality and pressure. Distribution system supplies the water from their source to usage point. Due to the leakage, the sufficient amount of water is not delivered to the consumer. Many researchers introduced the techniques for detecting the water leakage in distribution system. But, the water leakage detection accuracy was not improved and time consumption was also not reduced. To

improve the water leakage detection performance, Enhanced BrownBoost Classifier based Glowworm Swarm Optimization (EBBC-GWO) Method is introduced. EBBC-GWO method introduces two models namely, Enhanced BrownBoost Classifier model and Glowworm Swarm Optimization model. Enhanced BrownBoost Classifier model considers k-Nearest Neighbor (k-NN) classifier as weak classifier. It classifies the training samples with neighbor's majority vote for allocating the object to the class. Brownboost classifier combines all k-NN classifier to construct strong classifier. By this way, data are

classified as the normal data or abnormal data with higher accuracy. After classification, optimization process is executed where every solution corresponds to the glowworm (i.e., abnormal pressure data node) in search space. Every glowworm has objective function for addressing the optimization problem. Every glowworm operates in probabilistic means to choose the neighbor with higher luciferin value and transmit to it. Glowworm updates its location to the glowworm in dynamic decision space and optimal one is selected for water leakage detection. By this way, water leakage detection accuracy is improved

with lesser false positive rate. Experimental evaluation of proposed EBBC-GWO method is carried out with respect to number of pressure data and sensor placement nodes. The results demonstrated that EBBC-GWO method is higher in case of classification accuracy, false positive rate, classification time and water leakage detection accuracy. The simulation results show that EBBC-GWO method increases the performance of water leakage detection accuracy and reduces classification time when compared to state-of-the-art works

**1 Introduction**

Water Management System is efficient with development of automated system for leakage detection in water network (i.e., pipe break, water leakage, abnormal pressure or consumption). When the leaks are detected early, corrective actions are carried out to evade wastage of natural resources and economical losses. Leaks in water distribution network are detected through machine learning techniques. A fast water leakage detection system with adaptive design was introduced in



(Kang et al., 2018) that combined one-dimensional convolutional neural network and support vector machine (1D-CNN-SVM) model. A graph-based localization algorithm was introduced to find the leakage location. An actual water pipeline network was constructed in graph network and leakage occurred at virtual points on graph. However, the water leakage detection accuracy was not improved. A multi-stage graph partitioning approach was introduced in (Rajeswaran, Narasimhan and Narasimhan, 2018) to find the off-line flow measurements were required for reducing cost. The graph partitioning problem was addressed as the multi-objective mixed integer linear program (MILP). But, the classification performance was not carried out for water leak detection.

A new statistical framework was introduced in (Fagiani et al., 2016) for leakage identification. The framework comprises three sections for extraction and selection of features at leakage detection. A distinction of Sequential Feature Selection (SFS) algorithm was employed to choose the best performing features inside the set. But, the classification accuracy was not improved using new statistical framework. A wireless sensor network-based real-time monitoring system was introduced in (Kayaalp et al., 2017) to recognize and find leaks on water pipelines positions with pressure data. Three multi-label classification techniques like RA-kELd, BRkNN, and BR with SVM were employed for leak detection. A new leak detection technique was presented in (Sadeghioon et al., 2018) for water distribution pipelines as it buried pressurized fluid flow pipe. The detection technique was dependent on pressure sensor connected with temperature difference. An anomaly detection algorithm was introduced for monitoring the website traffic data that distinguishes the leak event from normal event. But, the false positive rate for water leakage detection was not reduced using anomaly detection algorithm.

The nominal variable was explained in (Dawidowicz, 2017) for addressing issues with pressure and division of water distribution system into the pressure zones. An artificial neural network was employed to address the problems raised in water distribution system. The classification was carried out depending on neural network variables explaining particular parameters that change the water distribution system design. A new model was introduced in (Dawidowicz, 2017) for leakage amount and location detection. A nodal pressures and demands were adjusted through multi-objective ant colony based optimization model. But, leakage detection time consumption was not reduced using ant colony based optimization model.

An algorithm structure was constructed in (Kumar et al., 2017) with modularity of wavelet and neural network that join the capability of wavelet transform through examining leakage signals and classification ability of artificial neural networks. The study authenticated that time domain was not apparent to features concerning noisy leak signals and significance of selection of mother wavelet in water distribution pipes. Leak detection was carried out in (Martini, Troncossi and Rivola, 2015) through vibration monitoring methods. The long-term objective was attained for automatic detection of leaks. An experimental model was computed for vibrations transmitted along water pipes through the burst leaks. The leak detection accuracy was not improved for automatic detection. The designed approach was introduced in (Sousa et al., 2015) depending on the steady-state modeling through monitoring tank flow and pressure at strategic nodes. The selection of pressure monitoring nodes was performed with graph theory ideas in WDN. The objective function was the reduction of



variation between estimated and measured pressures at the monitoring points. But, the classification process was not carried out in designed approach.

The certain problems are identified from the existing methods are lesser classification accuracy, higher classification time consumption, lesser water leakage detection accuracy, higher false positive rate, and so on. Therefore,
there is a necessity to design new efficient approach for water leakage detection.

The major contribution of the paper is described as follows,

- Enhanced BrownBoost Classifier based Glowworm Swarm Optimization (EBBC-GWO) Method is presented for optimal sensor placement to detect the water leakage with higher accuracy. EBBC-GWO method comprises two models namely, Enhanced BrownBoost Classifier model and Glowworm Swarm Optimization model.

- Enhanced BrownBoost Classifier model uses k-Nearest Neighbor (k-NN) classifier as a weak. k-NN classifier is a non-parametric method to classify training samples by majority vote of neighbor with object being assigned to the class. After that, Brownboost combines the weak learner to form the strong classifier. By this way, the data are classified as normal data or abnormal data with higher classification accuracy.

- In Glowworm Swarm Optimization model, every solution corresponds to the glowworm in the search space. Every
glowworm has the objective function for solving the optimization problem. Every glowworm sets the objective function at current location into luciferin value. Each glowworm functions in probabilistic manner to find the neighbor with higher luciferin value and travels toward it. Glowworm updates its location to the glowworm in dynamic decision space and optimal one is selected for water leakage detection.

The paper is structured as follows; Section 2 discusses the related works on water leakage detection. Section 3
explains Enhanced BrownBoost Classifier based Glowworm Swarm Optimization (EBBC-GWO) Method with neat diagram. Section 4 discusses the simulation setting. Section 5 presents result analysis of proposed EBBC-GWO method with various parameters. In section 6, the conclusion of the work is given.

## 2 Related works

Water Management System (WMS) is required by water utilities, municipalities and by corporate to address issues
with the water usage and supply. A new method was introduced in (Seyoum et al., 2017) for household leakage detection through sound signal recordings. The designed approach comprised recording of sound signals created through water fixtures and appliances. The recordings were employed to identify abnormal situation (i.e., leak). But, the classification accuracy was not improved. A new methodology was introduced in (Steffelbauer and Fuchs-Hanusch, 2016) with uncertainties of different sources in optimal sensor placement issues for leak localization on potential pressure measurement points. The issues were
addressed for diverse sensors and uncertainties. The sensor placement optimization issues were addressed in (Christodoulou et al., 2013) in urban water distribution networks through entropy-based approach for feasible water loss incident detection. The designed method was employed for longitudinal for acoustic, pressure or flow sensors on pipe segments. But, the optimization was not carried out in effective manner.



A new sensor placement approach was introduced in (Casillas et al., 2013) for leak location in water distribution networks (WDNs). The sensor placement issues were addressed through solving integer optimization problem. The optimization criterion reduced the number of non-isolable leaks consistent with the isolability criteria. However, sensor placement approach was not effective in detecting the water leak. Water leakage detection techniques were introduced and

issues were highlighted in (Adedeji et al., 2017). It accomplished the efforts and development in leakage detection technologies. It was efficient detection of background kind leakages. But, time consumption was not reduced using water leakage detection techniques. A leakage detection method was introduced in (Pérez et al., 2009) depending on discrepancies among pressure measurements and estimations attained from calibrated water distribution network. Every sensor in network identified discrepancy in pressure because of leakage based on its location.

A leak detection method was introduced in (Zhang and Wang, 2011) depending on Bayesian theory and Fisher's law for water distribution systems. A hydraulic model was linked with parameters of leaks. The randomness of parameter values was computed through the probability density function and Bayesian theory. The false positive rate was not lessened using leak detection method. A factorized distribution assigned the inflow demand across the consumption nodes with individual billing data and transmitted the demand across consumption nodes in (Moors et al., 2018) . The automatic leak

localization method with demand distribution models for computing the leak localization performance. But, the leak location identification is not carried out in effective manner.

A sampling design (SD) method was introduced in (Gamboa-Medina and Reis, 2017) for localization and quantification of pressure sensors in WDS for leak detection. SD addressed four criteria, namely improvement of leak sensitivity and sensitivity constancy and reduction of information redundancy and sensor count. However, the classification

time was not reduced by using SD method. A new technique was designed in (Islam et al., 2011) to identify and to diagnose the leakage in WDS. A fuzzy-based algorithm incorporated many uncertainties into many WDS parameters like roughness, nodal demands and water reservoir levels. Monitored pressure in nodes and flow in pipes computes the leakage possibility and its severity. But, the water leakage detection accuracy was not improved by fuzzy-based algorithm.

## 3 Methodology

Water leaks in Water Distribution Network leads to economic losses for final consumer in fluid transportation. In different WDN, losses because of leaks are calculated to 30% of extracted water. Water distribution is installed with underground pipes. The underground pipeline monitoring is difficult than open space water pipelines. Leaks in pipes are due to many several factors like pipe's age, improper fitting and disasters. A solution is needed to identify and to determine the location of damage in case of any leak. In order to detect the leak location, Enhanced BrownBoost Classifier based

Glowworm Swarm Optimization (EBBC-GWO) Method is introduced.



## 3.1 System Model

A water distributer system comprises the set of graph branches (i.e., $G = (V, E)$) where 'E' denotes set of edges (i.e., pipes) and 'V' denotes the set of vertices (i.e., nodes) explaining about the pipe connections and endings. A water distribution system is a collection of pipes (i.e., links) connected to the nodes (junctions, tanks and reservoirs). The water

5  flows are computed through hydraulic balancing, and through solving the equations at every node and links. EPANET software works out the network hydraulic equations automatically. The pressure and flow rate of every node and links are obtained from extended-period hydraulic simulation using EPANET software. EPANET software tool is simulated at without leak condition and pressure data readings are collected. The collected readings are taken as the threshold pressure data.

10  **3.2.2 Enhanced BrownBoost Classifier based Glowworm Swarm Optimization (EBBC-GWO) Method**

The key aim of EBBC-GWO method is to place the minimal number of sensors at optimal nodes for detecting the leakages. EBBC-GWO method comprises two models, namely Enhanced BrownBoost Classifier Model and Glowworm Swarm Optimization (GSO) Model for detecting the leakage in water distribution system.

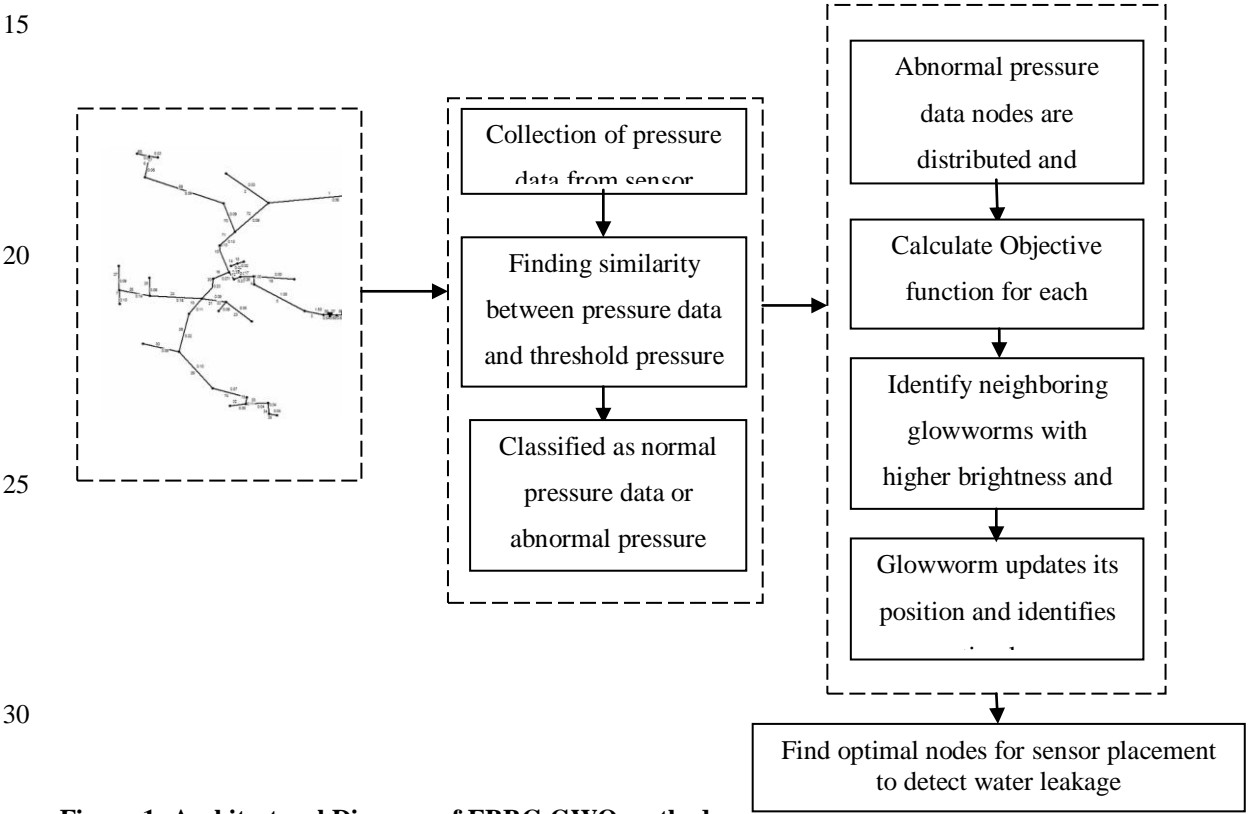

30

**Figure 1: Architectural Diagram of EBBC-GWO method.**

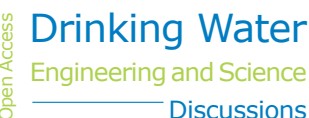

In Enhanced BrownBoost Classifier Model, the pressure data obtained from the EPANET tool is taken as an input. Then, the pressure data is classified as normal pressure data or abnormal pressure data with higher classification accuracy. After classification, abnormal pressure data is considered and the sensor is placed at the optimal nodes using Glowworm Swarm Optimization for detecting the leakage. The architectural diagram of EBBC-GWO method is illustrated in figure 1.

From figure 1, the architectural diagram of EBBC-GWO method is described. Initially, the water distribution network is constructed by using EPANET tool and reading is taken at without leak condition. After that, the pressure data are collected from the nodes in WDS. The similarity between the pressure data and threshold pressure data are calculated. The classification process is carried out as normal pressure data or abnormal pressure data. Then, abnormal pressure data nodes are distributed and taken as glowworms. The objective function of each pressure data is estimated. After that, the lesser

brighter glowworm moves towards the higher brighter glowworm and then updates its position. Finally, the optimal glowworms are taken for sensor placement to detect the water leakage. The brief explanation of Enhanced BrownBoost Classifier Model and Glowworm Swarm Optimization (GSO) Model are given next subsections

### 3.2.1 Enhanced BrownBoost Classifier Model

Initially, large number of pressure data is collected from nodes. After performing the data collection, classification

process is carried out by using Enhanced BrownBoost Classifier (EBBC) to find out the node is normal node or leakage possibility node. Enhanced BrownBoost classifier is a boosting method with two result outcomes. The traditional classification technique failed to classify the pressure data with more accuracy and minimal time for leakage detection. In order to increase the accuracy, Enhanced BrownBoost classifier is used which combines weak hypotheses (i.e. weak learner) to constructs the final strong classifier. EBBC-GWO method employs k-NN classifier as weak learner. k-Nearest Neighbor

(k-NN) classifier is non-parametric process to classify the training samples (i.e. pressure data of node) through majority neighbor votes for allocating to the class. In EBBC-GWO method, enhanced Brownboost classifier combines the weak learner to form the strong classifier. The flow process diagram of enhanced brownboost classifier is explained in figure 2.


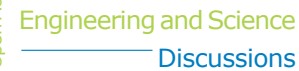



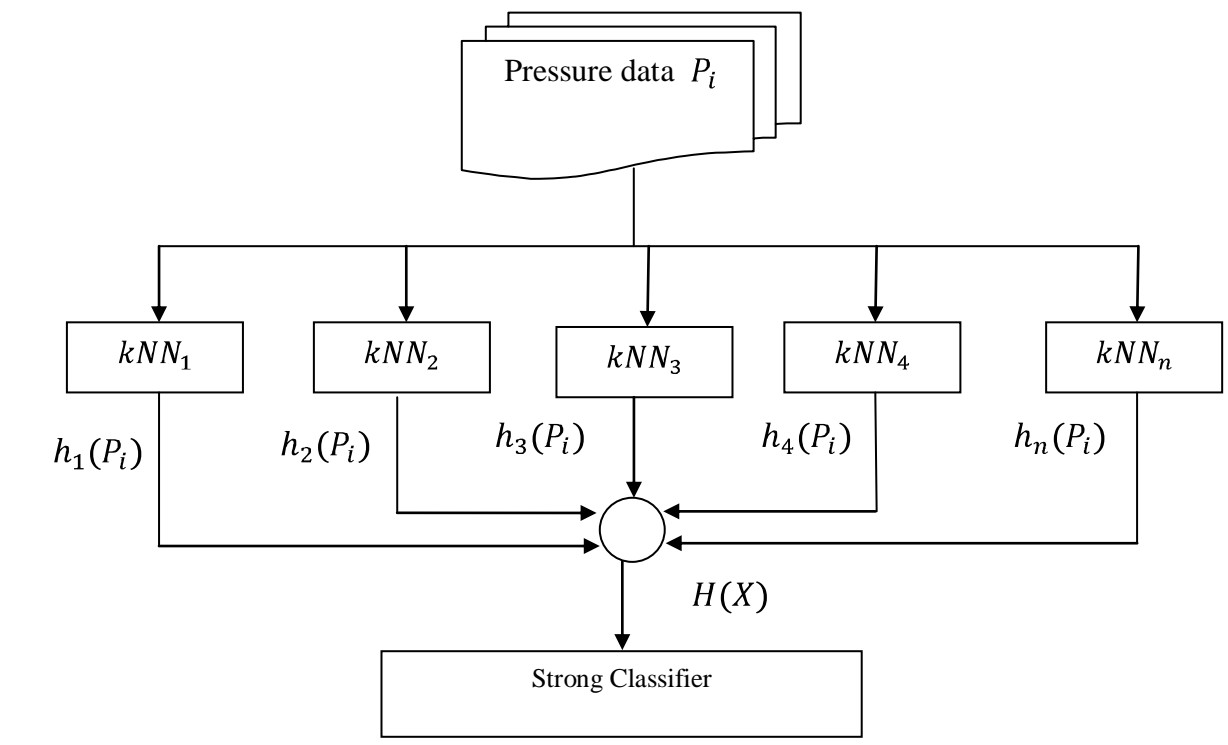

**Figure 2: Flow Process Diagram of Enhanced Brownboost Classifier.**

Figure 2 describes the enhanced BrownBoost Classification process with weak k-NN classifier for efficient classification. In enhanced BrownBoost classifier, k-NN base classifier is trained repeatedly with Pressure data. Enhanced BrownBoost classifier has 'n' number of training data $\{(P_1, y_1), (P_2, y_2), \dots (P_n, y_n)\}$ where '$P_i$' represents the set of pressure data and '$y_i$' denotes the final classifier output results. Let the threshold range of the pressure data be '$P_t$' (i.e., $P_t = P_{min} \leq P \leq P_{max}$). In

k-NN classifier, initially 'k' number of classes is initialized for classification. After initialization, the similarity between the training pressure data and threshold pressure range are to be computed in kNN classifier by using Extended Jaccard Similarity Measure. It is computed by,

$$h_j(x_i) = S(P_i, P_t) = \frac{P_i{}^T P_t}{\|P_i\|^2 + \|P_t\|^2 - P_i{}^T P_t} \tag{1}$$

From (1), the similarity measure '$h_j(x_i)$' is calculated and produces the output value as '0' or '1'. Based on the

similarity, the pressure data is classified as the normal data or abnormal data in kNN classifier. When the similarity output value is '0', then the pressure data is classified as abnormal data. When the similarity output value is '1', then the pressure data is classified as normal data. The base learner (i.e. kNN classifier) failed to classify the data in accurate manner. In order to increase the classification accuracy, brownboost classifier is employed. BrownBoost employs the non-convex potential loss function for classification. Consequently, initial potential loss function is given by

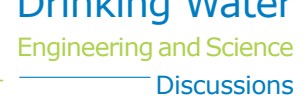

$$L_f = \frac{1}{N} \sum_{i=1}^{n} 1 - e_f(\sqrt{T}) \tag{2}$$

From (2), '$L_f$' represents the potential loss function of weak learner and 'N' denotes the number of pressure data, 'T' signifies the total amount of time taken by algorithm to run and '$e_f$' represents the error functions of each weak learner. The

sum of all potential loss function is calculated by,

$$L_f = \frac{1}{N} \sum_{j=1}^{K} \sum_{i=1}^{N} 1 - e_f \left( \frac{M_j(P_i)}{\sqrt{C}} \right) \tag{3}$$

$$L_f = 1 - e_f(\sqrt{C}) \tag{4}$$

From (3) and (4), '$M_j(p_i)$' represents the margin of weak learner of $i^{th}$ pressure data, 'C' represents positive real valued parameter. The final classifier is mixture of weak hypotheses. Initially, the weight of each weak learner (i.e., kNN classifier) is computed as,

$$w_i = e^{-\frac{(M_j(P_i) + T_r)^2}{c}} \tag{5}$$

From (5), '$w_i$' denotes weight of each weak learner, '$T_r$' represents the remaining amount of time. The weak learner is identified through the weight normalization. The value of '$\delta_i$' and '$t_r$' is computed by,

$$\sum_{j=1}^{K} \sum_{i=1}^{N} h_j(P_i) y_i e^{-\frac{(M_j(P_i) + \delta_i h_j(P_i) y_i + T_r - T)^2}{c}} = 0 \tag{6}$$

From (6), $\delta_i$ denotes an adjustment coefficient to ensure the final classification model. The margins for each

data is updated as,

$$M_j(p_{i+1}) = M_j(P_i) + \delta_i h_j(P_i) y_i \tag{7}$$

From (7), $M_j(p_{i+1})$ symbolizes the updated margins. The remaining time is updated as,

$$T_{r+1} = T_r - T \tag{8}$$

From (8), '$T_{r+1}$' denotes the remaining time, 'T' represents the variable amount of time related to weight given to the hypothesis. The output of strong classifier is given as,

$$H(X) = sign \left( \sum_{i=1}^{n} \delta_i h_j(P_i) \right) \tag{9}$$

From (9), '$H(X)$' represent the strong classifier output to increase classification accuracy. '$sign$' denotes the positive and negative outcomes of the classifier. The strong classifier is explained as,

$$H(X) = \begin{cases} +1 & \text{Normal pressure data} \\ -1 & \text{Abnormal pressure data} \end{cases} \tag{10}$$



From (10), $H(X) = +1$ represents the normal pressure data and $H(X) = -1$ symbolizes the abnormal pressure data. The algorithmic process of enhanced Brown boost classifier is described as follows.

---

\\ **Enhanced Brownboost Classifier Algorithm**

**Input**: Training data '$\{(P_1, Y_1), (P_2, Y_2), \dots (P_n, Y_n)\}$', remaining time '$T_r$', Adjustment coefficient '$\delta$'

**Output:** Increases classification accuracy and reduces classification time

**Step 1: Begin**

**Step 2:**     **For** each pressure data

**Step 3:**         Initialize $T_r$

**Step 4:**           Calculate $w_i$

**Step 5:**         Find weak classifier $\sum w_i h_i(x_i) y_i > 0$

**Step 6:**         Compute value of $\delta_i$ and $T_r$

**Step 7:**         Update the margins $M_j(p_{i+1})$

**Step 8**:         Update remaining time $T_{r+1}$

**Step 9**:         **if** $(H(X) = +1)$ **then**

**Step 10**:             Pressure data is normal

**Step 11:**         **else**

**Step 12:**             Pressure data is abnormal

**Step 13**:         **End if**

**Step 14**:     **End for**

**Step 15**: **End**

---

**Algorithm 1: Enhanced Brown boost Classifier Algorithm**

Algorithm 1 describes the enhanced Brownboost classifier with kNN base classifier for classifying the pressure data in efficient manner. Initially, the remaining time is initialized for performing the classification. Subsequently, the weak classifier is determined when output of weak learner is greater than zero. kNN classifier is combined for providing the better classification results. kNN classifier performs classification on votes of neighbor with object allocated to the class. After that, enhanced brownboost classifier combines the results of all kNN classifiers. The strong Brownboost classifier produces the results +1 or -1 where '+1' represents that the pressure data is normal and '-1' represents the pressure data is abnormal. Finally, enhanced Brownboost classifier improves the classification accuracy. After classification of pressure data, the abnormal pressure data are taken for performing the optimization process to place the sensor.

### 3.2.2 Glowworm Swarm Optimization (GSO) Model

The glowworm swarm optimization algorithm (GSO) is a kind of new swarm intelligence optimization method based on the behaviour of glowworms. The behaviour pattern of glowworms varies with the intensity of luciferin emission and glow with



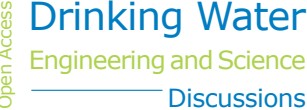

different intensities. The agents in GSO are taken as the glowworms (i.e, abnormal pressure data nodes) that send the luminescent called luciferin. Every glowworm uses luciferin (i.e., objective function) to transmit the information of the current location to their neighbors. Glowworms are attracted to brighter luciferin neighbors. The movement depends on the local information and selective neighbor interactions enable the swarm to partition into the disjoint subgroups that satisfies

the objective function.  In GSO algorithm, every solution corresponds to glowworm in search space and every glowworm has objective function (i.e., higher flow rate) for addressing the optimization problem (i.e., sensor placement on optimal node for leakage detection). Every glowworm sets the objective function at current location into luciferin value. The glowworms are based on the variable neighbourhood to detect the neighbors and compute the movements. Each glowworm employs the probabilistic way to choose the neighbor with higher luciferin value and moves toward it. Each glowworm has diverse

decision space with luciferin values enhanced than itself and distance inside decision radius. Glowworm renews the location to the glowworm in decision space probability. GSO has two main objectives:

- The agents glow at intensities proportional to objective function being optimized. It is taken that the glowworms of brighter intensities attracts the glowworms with lower intensity.

- GSO algorithm comprises decision range where distant glowworms is reduced when it has adequate neighbors

15       count or range beyond perception.

     Every glowworm comprises the local-decision domain bound by radial sensor range.  In local decision domain, every glowworm detects their neighbor and gets attracted by brighter glow of additional glowworms in neighbourhood set. Local-decision domain size is changeable and gets affected by its neighbors. If the neighbor has lesser density, local-decision domain increases for identifying the neighbors. Otherwise, local-decision domain gets reduced. Lastly, the movement of

glowworms resulted in the majority gathering to many optima. Figure 3 explains the directed graph among set of eight glowworms depending on relative luciferin levels and accessibility of local information.




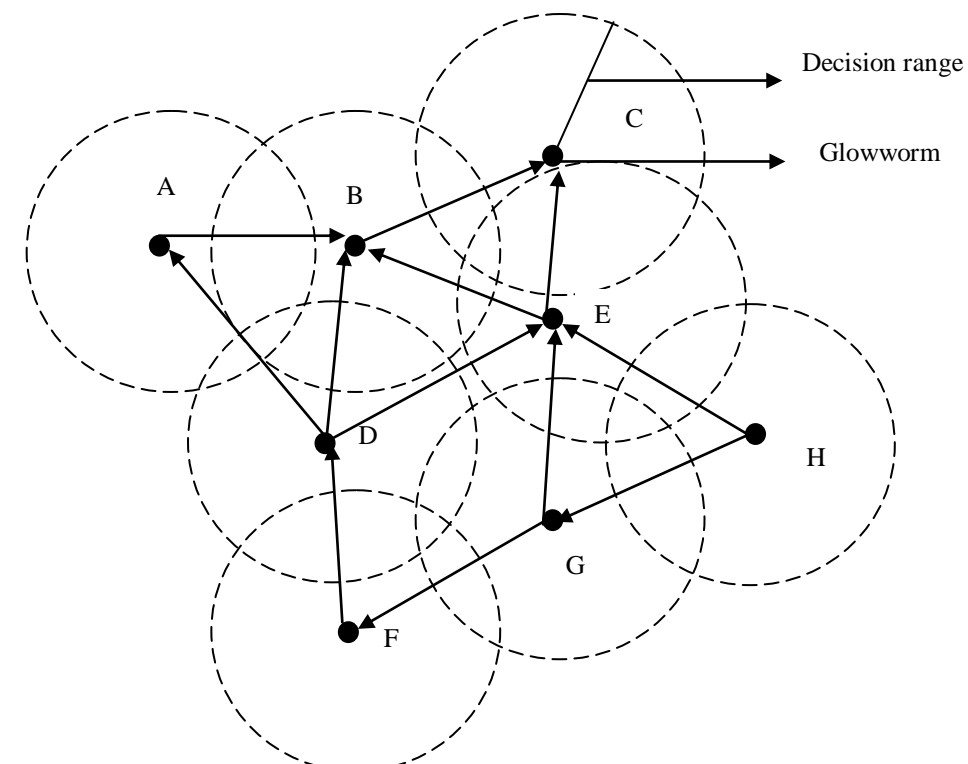

**Figure 3: Directed Graph of Eight Glowworms.**

The GSO algorithm comprises four steps, namely deployment of glowworms phase, luciferin-update phase, movement phase and local-decision domain update phase for finding the optimal glowworm.

**Deployment of glowworms phase:**

In this phase, the glowworms are distributed in whole objective space. Every glowworm comprises the equal quantity of luciferin and sensor range.

**Luciferin-update phase:**

The luciferin update is depending on the function value at glowworm position and all glowworms initiates the similar luciferin value during initial iteration. The values get varied consistent with the function values at current positions. During the luciferin update phase, every glowworm inserts to the previous luciferin level. A luciferin amount depends on the measured value of sensed profile. A fraction of luciferin value is subtracted to replicate the decay in the luciferin with time. Every glowworm varies luciferin value consistent with the objective function value of current location. The luciferin update rule is given by,

$$L_a(t+1) = (1-\rho)L_a(t) + \gamma J_a(t+1) \qquad (11)$$

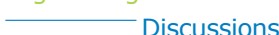

From (11), 'ρ' denotes the luciferin decay constant. '$L_a(t)$' symbolizes luciferin enhancement constant. '$J_a(t)$' represent the objective function value at glowworm '$a^{th}$' location at time 't'. '

**Movement-phase**

During the movement-phase, each glowworm takes the decision by probabilistic method for shifting to their neighbor with higher luciferin value. They are attracted to the neighbors which glow brighter. For every glowworm 'a', probability of moving to the neighbor 'b' is formulated,

$$N_a(t) = \{b: D_{ab}(t) < R_d{}^a(t)\}; L_a(t) = L_b(t) \qquad (12)$$

$$\rho_{ab}(t) = \frac{L_b(t) - L_a(t)}{\sum_{k \in N_a(t)} L_b(t) - L_a(t)} \qquad (13)$$

From (12) and (13), 't' denotes the time index. '$D_{ab}(t)$' signify the Euclidian distance between glowworms 'a' and 'b' at time 't'. $L_b(t)$ represent the luciferin level with glowworm 'b' at time 't'. '$R_d{}^a(t)$' denotes the variable local-decision range linked with glowworm 'a' at time 't'. 'R' is the radial range of luciferin sensor. Each glowworm updates their location by,

$$X_a(t+1) = X_a(t) + s\left(\frac{X_b(t) - X_a(t)}{\|X_b(t) - X_a(t)\|}\right) \qquad (14)$$

From (14), 's' denotes the step size. '$\|X_b(t) - X_a(t)\|$' symbolizes the Euclidean Operator of glowworms.

**Local-decision range update rule**

When glowworms are depending on the local information to decide movements, it is considered that the number of peaks captured is strong function of radial sensor range. The radius of every glowworm dynamic decision space is utilizes the current radius of dynamic decision space and links with the radial range of luciferin sensor. The update rule of every glowworm dynamic decision space radius is given by,

$$R_d{}^a(t+1) = \min\{R, \max\{0, R_d{}^a(t) + \beta(N_t - |N_a(t)|\}\} \qquad (15)$$

From (15), 'β' denotes the constant parameter. '$N_t$' symbolizes explicit threshold parameter. From this updated rule, the best glowworm is identified. The algorithmic process of Glowworm Swarm Optimization is given as follows,



---

\\ **Glowworm Swarm Optimization Algorithm**

**Input**: Abnormal pressure data nodes

**Output**: Optimizes the node for sensor placement

**Step 1**: Begin

**Step 2**:      For each abnormal pressure data nodes

**Step 3**:           Calculate the objective function

**Step 4**:            Find probability of movement by comparing objective function value with its neighbor

**Step 5**:          if (Neighbor node objective function > current node)  then

**Step 6**:               Movement takes place to neighbouring position

**Step 7**:          else

**Step 8**:               No movement takes place

**Step 9**:           End if

**Step 10**:           Updates position of node

**Step 11**:      End for

**Step 12**:  Selects optimal node for sensor placement

**Step 13**: End

---

**Algorithm 2: Glowworm Swarm Optimization Algorithm**

Algorithm 2 explains the Glowworm Swarm Optimization for selecting the optimal one. The agents in GSO are the
abnormal pressure data nodes. Every glowworm estimates their objective function and sends the information of current
location to their neighbors based on objective function. Lesser objective function nodes get attracted by brighter objective
function neighbors. Every glowworm takes the decision through probabilistic mechanism for moving to its neighbor with
higher objective function. After that, the position of nodes gets updated and the optimal node is selected for the sensor
placement to detect the leakage in WDS. By this way, the water leakage detection accuracy is improved.

**4 Simulation Settings**

Enhanced BrownBoost Classifier based Glowworm Swarm Optimization (EBBC-GWO) Method is implemented in
MATLAB Simulink with 3.4 GHz Intel Core i3 processor, 4GB RAM, and windows 7 platform for water leak detection. The



nodes pressure and demand flow in water pipeline networks are collected at without leak condition from extended-period hydraulic simulation using EPANET software. EBBC-GWO method is used for District Metered Areas (DMA) in Barcelona water distribution network. Total extent of DMA is 17.4 km of pipelines with 883 nodes, 927 pipes which distributes water to 639 consumers. The pipe diameters varied from 70 to 400 mm. The pressure at night flow changes between 29.31m and

43.46 m. Flow and pressure are identified at inflow and outflow point. DMA includes 311 nodes with demand (RM), 60l nodes without demand (EC), 48 hydrant nodes without demand (HI type), 14 dummy valve nodes without demand (VT) and 448 dummy nodes without demand (XX type). The water pipeline has two inflow inputs (i.e., reservoir nodes). Leak detection is based on the damage (leaks) in different locations of piping network includes the liquid outflow at leak location that changes the flow features (pressure heads, flow rates, acoustics signals, etc.) at monitoring locations of piping network.

It is imagined that leaks occur at XX type nodes in which 448 potential leaks are detected. Leaks occur at any node or pipe. Flow rate are employed for leak detection and gathering pressure data.

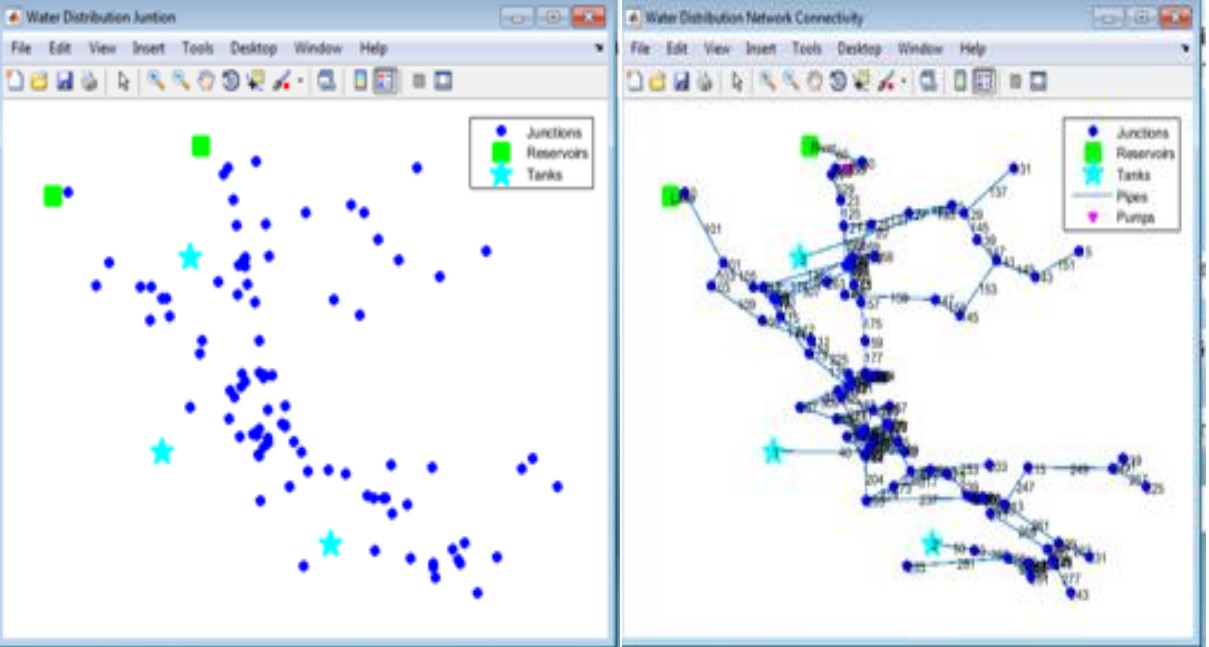

**Figure 4: Simulation Diagram of Water Distribution Network.**

## 5  Simulation Result Analysis

Enhanced BrownBoost Classifier based Glowworm Swarm Optimization (EBBC-GWO) Method is designed for detecting the water leakage in WDS and compared with existing one-dimensional convolutional neural network and support vector machine (1D-CNN-SVM) model (Kang et al., 2018)  and Multi-Stage Graph Partitioning Approach (Rajeswaran, Narasimhan and Narasimhan, 2018) . The efficiency of EBBC-GWO method is evaluated along with the metrics such as classification accuracy, classification time, water leakage detection accuracy and false positive rate.



## 5.1 Classification Accuracy (CA)

Classification accuracy is measured as the ratio of number of pressure data that are correctly classified to the total number of pressure data. It is measured in terms of percentage (%). It is given by,

$$CA = \frac{Number\ of\ pressure\ data\ correctly\ classifed}{Total\ number\ of\ pressure\ data} * 100 \qquad (16)$$

5    **Sample calculation:**

➤ **1D-CNN-SVM model:** the number of pressure data correctly classified is 30 and the total number of pressure data is 50.  After that, classification accuracy is computed as,

$$CA = \frac{30}{50} * 100 = 60\ \%$$

➤ **Multi-Stage Graph Partitioning:** the number of pressure data correctly classified is 39 and the total number of pressure data is 50.  After that classification accuracy is calculated as,

$$CA = \frac{39}{50} * 100 = 78\ \%$$

10   ➤ **Proposed EBBC-GWO Method:** the number of pressure data correctly classified is 45 and the total number of pressure data is 50. After that, the classification accuracy is obtained as,

$$CA = \frac{45}{50} * 100 = 90\ \%$$

When the classification accuracy is higher, method is efficient.

**Table 1: Tabulation for Classification Accuracy**

| Number of Pressure data (Number) | Classification Accuracy (%) | | |
| --- | --- | --- | --- |
| | 1D-CNN-SVM model | Multi-Stage Graph Partitioning Approach | EBBC-GWO Method |
| 50 | 60 | 78 | 90 |
| 100 | 58 | 80 | 96 |
| 150 | 67 | 82 | 90 |
| 200 | 68 | 82 | 93 |
| 250 | 80 | 90 | 98 |
| 300 | 80 | 92 | 96 |
| 350 | 80 | 85 | 94 |
| 400 | 85 | 91 | 98 |
| 450 | 83 | 89 | 96 |
| 500 | 88 | 93 | 98 |



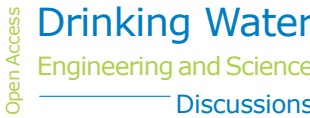

Table 1 describes classification accuracy of three different methods implemented in Matlab simulation by taking diverse number of pressure data ranging from 50-500. The simulation result of classification accuracy using EBBC-GWO Method is compared with existing 1D-CNN-SVM model (Kang et al., 2018) and Multi-Stage Graph Partitioning Approach (Rajeswaran, Narasimhan and Narasimhan, 2018) . From the table, it clear that the classification accuracy using EBBC-GWO Method is higher than other existing techniques (Kang et al., 2018), (Rajeswaran, Narasimhan and Narasimhan, 2018) . The performance result of classification accuracy is illustrated in figure 5.

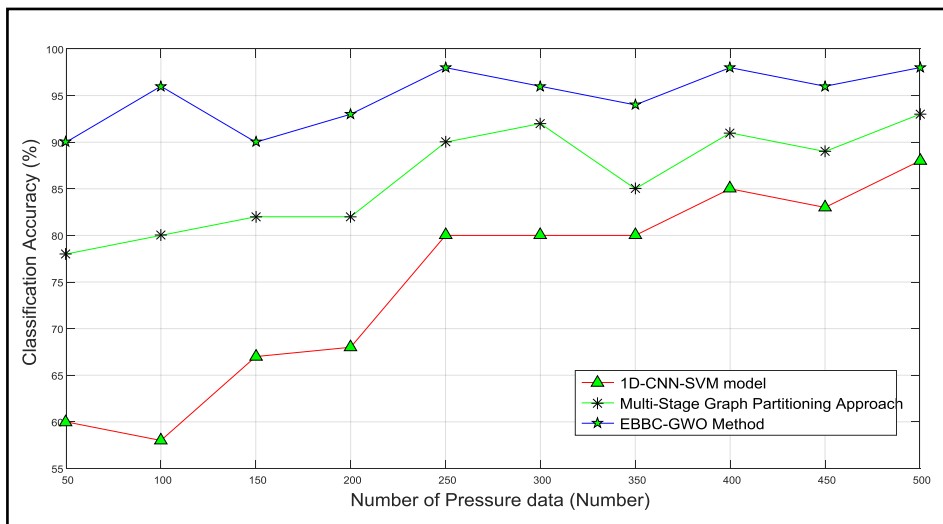

**Figure 5: Measurement of Classification Accuracy**

Figure 5 illustrates the classification accuracy performance for different number of pressure data. In above graph, different number of pressure data is taken in 'X' axis and the classification accuracy is taken in 'Y' axis. From figure, the blue color line represents the classification accuracy of EBBC-GWO Method where green color line and red color line represents the classification accuracy of PF 1D-CNN-SVM model (Kang et al., 2018) and Multi-Stage Graph Partitioning Approach.

Classification process is performed to identify the data as normal data or abnormal data to further processing. For performing the classification, enhanced brownboost classifier is used in our work. In this classifier, k-nearest neighbor classifier is taken as base classifier. The enhanced brownboost classifier combines the output of all classifier to produce strong classifier with higher classification accuracy. In our work, the number of pressure data is taken as training samples. After that, k-nearest neighbor classifier classifies the pressure data as normal data or abnormal data through finding the similarity between the pressure data and threshold pressure data. After obtaining the results from all weak classifier, enhanced brownboost classifier boosts the results through combining all kNN classifiers to produce the strong classifier output. By this way, the classification of pressure data is carried out with higher accuracy.

Let us consider the number of instances as 10. Every instance the number of pressure data gets varied. When number of pressure data is taken as 300 for simulation work, EBBC-GWO Method gets 96% of classification accuracy while



1D-CNN-SVM model (Kang et al., 2018) and Multi-Stage Graph Partitioning Approach (Rajeswaran, Narasimhan and Narasimhan, 2018) gets 80% and 92% of classification accuracy respectively. The classification accuracy of EBBC-GWO Method is increased by 29% and 10% compared to existing 1D-CNN-SVM model (Kang et al., 2018) and Multi-Stage Graph Partitioning Approach (Rajeswaran, Narasimhan and Narasimhan, 2018) respectively.

**5.2 Classification Time**

Classification time is defined as amount of time taken to classify the data as normal data and abnormal data for optimal sensor placement at the nodes in the water distribution network. It is the difference of starting and ending time of classification. It is measured in terms of milliseconds (ms). It is formulated as

$$CT = Ending\ time - Strating\ time\ of\ classification \qquad (17)$$

**Sample calculation:**

➢ **1D-CNN-SVM model:** the starting time of classification is 0 and the ending time of classification is 72. After that, classification time is calculated as,

$$CT = 72 - 0 = 72ms$$

➢ **Multi-Stage Graph Partitioning:** the starting time of classification is 0 and the ending time of classification is 52. After that classification time is calculated as,

$$CT = 52 - 0 = 52ms$$

➢ **Proposed EBBC-GWO Method:** the starting time of classification is 0 and the ending time of classification is 36. After that, the classification time is obtained as,

$$CT = 36 - 0 = 36ms$$

When the classification time is lesser, method is more efficient.



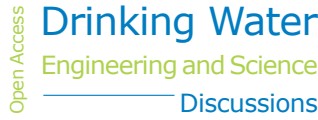

**Table 2: Tabulation for Classification Time**

| Number of Pressure Data (Number) | Classification Time (ms) | | |
|---|---|---|---|
| | 1D-CNN-SVM model | Multi-Stage Graph Partitioning Approach | EBBC-GWO Method |
| 50 | 72 | 52 | 36 |
| 100 | 76 | 45 | 40 |
| 150 | 70 | 56 | 43 |
| 200 | 79 | 62 | 52 |
| 250 | 85 | 68 | 46 |
| 300 | 91 | 60 | 58 |
| 350 | 98 | 71 | 62 |
| 400 | 106 | 74 | 55 |
| 450 | 99 | 85 | 66 |
| 500 | 118 | 93 | 75 |

Table 2 describes classification time of three different techniques implemented in Matlab simulation with number of pressure data ranging from 50-500. The performance of classification time using EBBC-GWO Method is compared with the existing 1D-CNN-SVM model (Kang et al., 2018) and Multi-Stage Graph Partitioning Approach (Rajeswaran, Narasimhan and Narasimhan, 2018) . From the table, it clear that the classification time using EBBC-GWO Method is lesser than other existing techniques (Kang et al., 2018), (Rajeswaran, Narasimhan and Narasimhan, 2018) . The simulation graph of classification time is described in figure 6.



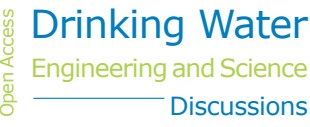

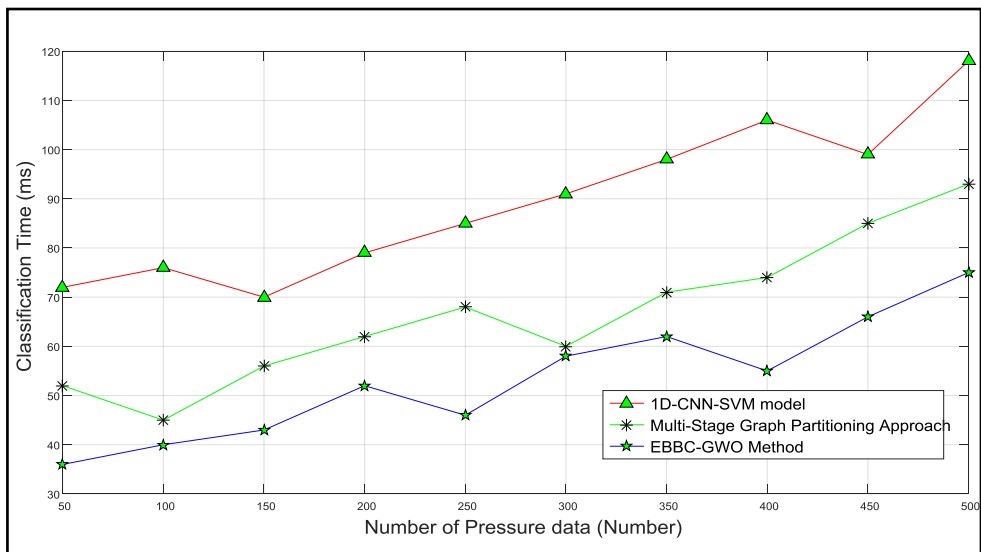

**Figure 6: Measurement of Classification Time**

In figure 6, the classification time comparison is carried out for different number of pressure data. In the above graph, number of pressure data is taken in 'X' axis and the classification time is taken in 'Y' axis. From the figure, the blue color line represents the classification time of EBBC-GWO Method where green color line and red color line represents the classification time of 1D-CNN-SVM model (Kang et al., 2018) and Multi-Stage Graph Partitioning Approach (Rajeswaran, Narasimhan and Narasimhan, 2018) .

Classification process categorizes the training data as the normal data or abnormal data for future use. An enhanced brownboost classifier is employed for the classification process in our research work. In this classifier, training samples are given for all k-nearest neighbor classifier at the same time to reduce the time consumption for classification. After that, the output from all the weak classifier is combined to produce the strong classifier. In our work, the number of pressure data is taken as input for all weak classifiers at the same time. After that, k-nearest neighbor classifier categorizes the pressure data as normal or abnormal data through discovering the extended jaccard similarity between the pressure data and threshold pressure data. After obtaining the results from all weak classifier, enhanced brownboost classifier improves the results through combining all kNN classifiers to form the strong classifier. By this way, the classification of pressure data is carried out with minimal amount of time.

Let us take the number of instances as 10. For every instance, the number of pressure data gets changed. When number of pressure data is taken as 200, EBBC-GWO Method classifies the pressure data in 52ms while 1D-CNN-SVM model (Kang et al., 2018) and Multi-Stage Graph Partitioning Approach (Rajeswaran, Narasimhan and Narasimhan, 2018) classifies the data in 79ms and 62ms respectively. The classification time consumption of EBBC-GWO Method is decreased by 41% and 20% compared to existing 1D-CNN-SVM model (Kang et al., 2018) and Multi-Stage Graph Partitioning Approach (Rajeswaran, Narasimhan and Narasimhan, 2018) respectively.





### 5.3 Water Leakage Detection Accuracy (WLDA)

Water leakage detection accuracy is the process of detecting the leak accurately through placing the sensor at optimal nodes. It is measured in terms of percentage (%). It is measured by,

$$WLDA = \frac{Number\ of\ sensor\ nodes\ that\ detects\ leak\ accurately}{Number\ of\ sensor\ placement\ nodes} * 100 \qquad (18)$$

**Sample calculation:**

➢ **1D-CNN-SVM model:** the number of sensor nodes that detects leak accurately is 35 and the number of sensor placement nodes is 50. After that, water leakage detection accuracy is computed as,

$$WLDA = \frac{35}{50} * 100 = 70\%$$

➢ **Multi-Stage Graph Partitioning:** the number of pressure data correctly classified is 39 and the total number of

10      sensor placement nodes is 50. After that water leakage detection accuracy is calculated as,

$$WLDA = \frac{39}{50} * 100 = 79\%$$

➢ **Proposed EBBC-GWO Method:** the number of pressure data correctly classified is 43 and the total number of sensor placement nodes is 50. After that, the water leakage detection accuracy is obtained as,

$$WLDA = \frac{43}{50} * 100 = 86\%$$

When the water leakage detection accuracy is higher, the method is efficient.

**Table 3: Tabulation for Water Leakage Detection Accuracy**

| Number of Sensor Placement Nodes (Number) | Water Leakage Detection Accuracy (%) | | |
|---|---|---|---|
| | 1D-CNN-SVM model | Multi-Stage Graph Partitioning Approach | EBBC-GWO Method |
| 50 | 70 | 78 | 86 |
| 100 | 63 | 80 | 90 |
| 150 | 70 | 75 | 90 |
| 200 | 75 | 81 | 88 |
| 250 | 79 | 84 | 88 |
| 300 | 72 | 88 | 95 |
| 350 | 82 | 89 | 97 |
| 400 | 81 | 91 | 97 |
| 450 | 89 | 91 | 97 |
| 500 | 90 | 93 | 98 |



Table 3 explains the water leakage detection accuracy of three different techniques with respect to number of sensor placement nodes ranging from 50-500. The performance of water leakage detection accuracy using EBBC-GWO Method is compared with the existing 1D-CNN-SVM model (Kang et al., 2018) and Multi-Stage Graph Partitioning Approach

(Rajeswaran, Narasimhan and Narasimhan, 2018). From the table, it is clear that the water leakage detection accuracy of EBBC-GWO Method is higher than other existing techniques (Kang et al., 2018), (Rajeswaran, Narasimhan and Narasimhan, 2018) . The simulation graph of water leakage detection accuracy is described in figure 7.

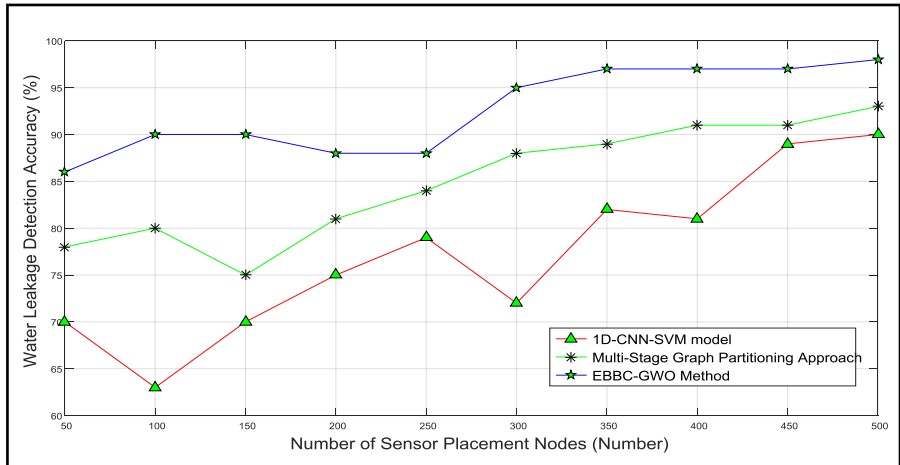

**Figure 7: Measurement of Water Leakage Detection Accuracy**

In figure 7, the water leakage detection accuracy comparison is carried out for different number of sensor placement nodes. In above graph, number of sensor placement nodes is taken in 'X' axis and the water leakage detection accuracy is taken in 'Y' axis. From the figure, the blue color line denotes the water leakage detection accuracy of EBBC-GWO Method where green color line and red color line denotes the water leakage detection accuracy of 1D-CNN-SVM model (Kang et al., 2018)  and Multi-Stage Graph Partitioning Approach (Rajeswaran, Narasimhan and Narasimhan, 2018) .

Classification process classifies the training data as the normal data or abnormal data. After performing the classification process, optimization process is carried out to identify the optimal node. An enhanced brownboost classifier is employed for the classification process and glowworm swarm optimization is employed for choosing the optimal one in our research work. In this classifier, the number of pressure data is taken as input. K-nearest neighbor classifier classifies the pressure data as normal or abnormal data through finding the extended jaccard similarity between pressure data and

threshold pressure data. Then, enhanced brownboost classifier combines all kNN classifiers output to form the strong classifier. After that, the abnormal pressure data node is taken and objective function is calculated for all data. Based on the brightness movement takes place and position gets updated. Finally, the optimal node is selected based on the brightness and water leakage gets detected. By this way, the water leakage gets detected with higher accuracy.



Let us take the number of instances as 10. For every instance, the number of sensor placement nodes gets changed. When number of sensor placement data is taken as 400, EBBC-GWO Method obtains the 97% of water leakage detection accuracy while 1D-CNN-SVM model (Kang et al., 2018) and Multi-Stage Graph Partitioning Approach (Rajeswaran, Narasimhan and Narasimhan, 2018) produces 81% and 91% of water leakage detection accuracy respectively. The water

leakage detection accuracy of EBBC-GWO Method is increased by 21% and 9% compared to existing 1D-CNN-SVM model (Kang et al., 2018) and Multi-Stage Graph Partitioning Approach (Rajeswaran, Narasimhan and Narasimhan, 2018) respectively.

### 5.4 False Positive Rate

False positive rate is defined as the leak that is incorrectly detected by the sensor placed nodes. It is measured in

terms of percentage (%). False positive rate is formulated as,

$$False\ Positive\ Rate = \frac{Number\ of\ leaks\ incorrectly\ identified\ by\ sensor}{Number\ of\ sensor\ placement\ nodes} \qquad (19)$$

**Sample calculation:**

➤   **1D-CNN-SVM model:** the number of sensor nodes that incorrectly detects leak is 15 and the number of sensor

placement nodes is 50. After that, false positive rate is calculated as,

$$WLDA = \frac{15}{50} * 100 = 30\%$$

➤   **Multi-Stage Graph Partitioning:** the number of sensor nodes that incorrectly detects leak is 11 and the total number of pressure data is 50. After that false positive rate is calculated as,

$$WLDA = \frac{11}{50} * 100 = 22\%$$

➤   **Proposed EBBC-GWO Method:** the number of sensor nodes that incorrectly detects leak is 7 and the total number of pressure data is 50. After that, the false positive rate is obtained as,

$$WLDA = \frac{7}{50} * 100 = 14\%$$

When the false positive rate is lesser, the method is said to be more efficient.

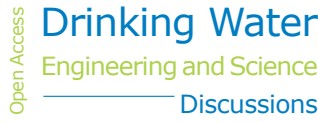

**Table 4: Tabulation for False Positive Rate**

| Number of Sensor Placement Nodes (Number) | False Positive Rate (%) | | |
|---|---|---|---|
| | 1D-CNN-SVM model | Multi-Stage Graph Partitioning Approach | EBBC-GWO Method |
| 50 | 30 | 22 | 14 |
| 100 | 37 | 20 | 10 |
| 150 | 30 | 25 | 10 |
| 200 | 26 | 20 | 13 |
| 250 | 21 | 16 | 12 |
| 300 | 28 | 12 | 5 |
| 350 | 18 | 11 | 3 |
| 400 | 19 | 9 | 3 |
| 450 | 11 | 9 | 3 |
| 500 | 10 | 7 | 2 |

Table 4 describes the false positive rate of three different techniques for number of sensor placement nodes ranging from 50-500. The performance of false positive rate using EBBC-GWO Method is compared with the existing 1D-CNN-SVM model (Kang et al., 2018) and Multi-Stage Graph Partitioning Approach (Rajeswaran, Narasimhan and Narasimhan, 2018). From the table, it is clear that the false positive rate of EBBC-GWO Method is lesser than other existing techniques (Kang et al., 2018), (Rajeswaran, Narasimhan and Narasimhan, 2018). The simulation graph of false positive rate is explained in figure 8.

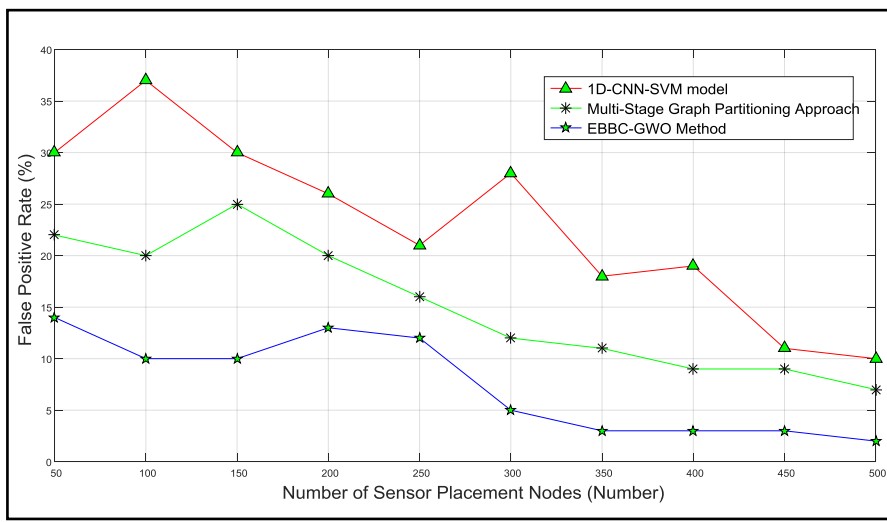

**Figure 8: Measurement of False Positive Rate**

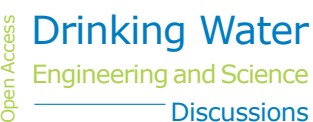

Figure 8 describes the false positive rate comparison for different number of sensor placement nodes. From the graph, number of sensor placement nodes is taken in 'X' axis and the false positive rate is taken in 'Y' axis. In figure, the blue color line symbolizes the false positive rate of EBBC-GWO Method where green color line and red color line represents the false positive rate of 1D-CNN-SVM model (Kang et al., 2018) and Multi-Stage Graph Partitioning Approach (Rajeswaran,

Narasimhan and Narasimhan, 2018) .

Classification process is carried out to classify the training data as the normal data or abnormal data for performing the optimization process. An optimization process is carried out to identify the data as optimal one. An enhanced brownboost classifier is classifies the data and glowworm swarm optimization optimizes the data to detect the water leakage in our research work. In this classifier, the number of pressure data is taken as input. Enhanced brownboost classifier combines all

kNN classifiers results to form the strong classifier output. Then, the abnormal pressure data node is distributed and objective function is computed for all data. Depending on the objective node, the optimal nodes are selected. Finally, water leakage gets detected by placing the sensor at that particular node. By this way, the false positive rate gets reduced in our research work.

For conducting the simulation experiment, number of instances taken are 10. For every instance, the number of

sensor placement nodes gets changed. When number of sensor placement data is taken as 350, EBBC-GWO Method attains the 3% of false positive rate while 1D-CNN-SVM model (Kang et al., 2018) and Multi-Stage Graph Partitioning Approach (Rajeswaran, Narasimhan and Narasimhan, 2018) produces 18% and 11% of false positive rate respectively. The false positive rate of EBBC-GWO Method is reduced by 69% and 54% compared to existing 1D-CNN-SVM model (Kang et al., 2018) and Multi-Stage Graph Partitioning Approach (Rajeswaran, Narasimhan and Narasimhan, 2018) respectively.

**6 Conclusion**

An Enhanced BrownBoost Classifier based Glowworm Swarm Optimization (EBBC-GWO) Method is introduced for leakage detection in water distribution system. Enhanced BrownBoost Classifier model taken k-Nearest Neighbor (k-NN) classifier as weak classifier and classifies training samples through majority vote of neighbor. After that, Brownboost classifier combines all k-NN classifier output to build the strong classifier. By this way, data are classified as the normal data

or abnormal. After classification, glowworm swarm optimization process is carried out where every solution corresponds to abnormal pressure data node in search space. Every glowworm has objective function for addressing the optimization issue. Every glowworm employs probabilistic way to choose neighbor with higher objective function value and moves toward it. Glowworm updates its location to the glowworm in dynamic decision space and optimal one is selected for senor placement to detect the leakage. By this way, water leakage detection accuracy is improved with lesser false positive rate. Experimental

evaluation is carried out with the parameters such as classification accuracy, false positive rate, classification time and water leakage detection accuracy. The results analysis of EBBC-GWO method improves classification accuracy and water leakage detection accuracy with minimum false positive rate and classification time than the state-of-art methods.



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
