# Peer review of "Hybridisation of brownboost classifier and glowworm swarm based optimal sensor placement for water leakage detection"

_Drinking Water Engineering and Science, 2018_

## Referee Comment (RC1) · Anonymous Referee #1 · 21 Nov 2018

In the paper a new methodology is introduced to localize leakages in distribution networks. It is extensive work and the results of the simulations are compared to two other methods. General comments: - The abstract is too long, and should not focus too much on the description of the methodology - Shorten introduction and make the description of the various methodologies more condensed at analyse the short-comings at once, so that it gives a good reason for the adopted approach. - Introduction and "related work" should be combined, because in the introduction related work should already be cited to set the boundaries for the proposed approach. - Mention at the end of the

introduction what is the reason to develop the proposed approach (and what is new). - Avoid in the methodology section using s "code" and too detailed descriptions in the main text (preferably in supplementary data/information), but concentrate more on how the calibration/validations are done. Are real pressure data used or is it all simulated data, e.g.? - In the results section new definitions should be avoided (should be introduced in Methodology section). Also avoid repetition and figures and tables with same information. It is suggested to present one large table with all data in it and then discuss the differences of the three methods at once. - It is not clear from the results how the localization of the sensors is determined - It should be good to discuss the implementation of the proposed approach in practice. Is it for example needed to run the EPAnet model continuously to detect anomalies? - Check tenses: past tense when it is the author's work, present tense when it is general knowledge - Check abbreviations: introduce once and then always use abbreviation (not introducing again) - Check language in general.

Specific comments: - Page 1, Line 26, unclear what is meant - Page 1, line 27 give references - Page 1, line 28 evade=avoid - Page 2, line 1 "..in (kang et al. ...) that.." should be something like "Kang et al. (2018) found..." check rest of the document too. - Page 2, line 6-7 stick to references that are related to leak detection in pressurized pipes. - Page 2, line 33 introduce abbreviation - Page 3, line 5-18, is more for the methodology section - Page 4, line 26, give references (not always the case...) - Page 9, line 4-5 this statement should be proven by the results (and is not adequate here). - Page 14, Figure 4 is not readable - Page 15, line 2 Explain better (in methodology section) what is meant by "classified" - Page 24, line 29-31 Not clear what is meant... Only less false positive or also better on other 3 performance indicators? What is meant by "experimental evaluation"? What experiments are executed?
* * *

---

## Referee Comment (RC2) · Anonymous Referee #2 · 21 Nov 2018

General comments:

The manuscript presents a method to identify optimal pressure measurement positions in a water distribution system for leak detection by utilising classification and meta-heuristic optimisation algorithms. However, the proposed method is questionable and the scientific style how this method is presented in this paper is insufficient. Besides the manuscript being poorly readable, the article lacks necessary technical details crucial for understanding the methodology as well as references to related key literature. Furthermore, some parts in the methods section are missing references at all, resulting

in plagiarism (intended or not), especially in the parts describing brownboost classifiers and glowworm swarm optimisation—the key methods of this paper. The paper contains 8 full pages (pages 6 to 13) copying ideas from other authors without citing them adequately. Even without these concerns, the novelty and usefulness of the proposed methods is arguable. In my opinion, the only novelty lies in the authors applying very specific algorithms (brownboost classifier, glowworm swarm) on a problem that has been already addressed in former literature with various other classification and optimisation algorithms. The results—recommending to put 50 to 500 sensors in a network of 17 km (up to a sensor every 34 (!) meter)—are without doubt unfeasible for applying the methodology on real-world systems. In conclusion, the reviewer could not recommend the manuscript for publication in DWES.

Nevertheless, the authors are invited to consider the following more specific comments:

Specific comments:

Introduction and Related works:

-The introduction is actually missing an introduction: Why is finding leaks important? How do water utilities find leaks (see for example Puust et al. 2010)? What are classical methods, what are recent methods? What are benefits of model-based approaches? After a few incoherent introductory sentences, the introduction lists scientific literature at random. For the reader it is impossible to find any reasoning behind the order of the references listed in the paper.

-Page 1 Line 28: The sentence "leaks in WDN are detected through machine learning techniques" is not true, these methods are just one way to detect leaks, in fact, used only in scientific literature and not in practice at all.

-In general, the listed literature is a mix of different methods for leak detection, which are even focussing on different physical effects caused by a leak (e.g. flow, noise, pressure without even mentioning it here) as well as a wild mix between leak localisation,

leak detection and optimal sensor placement methods. Therefore, they can't be compared and listed in the way they are presented in introduction and the related works section.

-The literature review is insufficient: Only very recent literature is listed—the oldest publication is from 2015. Older key literature as well as novel key literature for this topic is missing. In my opinion, the state of the art is not well described. I recommend the authors to review the literature in the papers that the mention and identify the key papers for this topic.

-The knowledge gap is not well defined. There are only sentences dropped in the introduction like "however, XXX was not improved", "but YYY was not carried out" or "ZZZ was not reduced" without stating if that was even the purpose or aim of the listed paper and without giving any further explanations.

-Page 3 Line 8: Throughout the paper it is mentioned that the method allows to detect leaks with higher accuracy, but it is not clear what is meant. Higher than what?

-Page 3 Line 20: with "neat" diagram sounds strange.

-Page 3 Line 23: Related works: Is the work listed in the introduction not also related work? What is the difference of this section to the previous one?

-Page 3 Line 13: The paper states the difference between normal data and abnormal data, but it is not clear throughout the paper what this terms actually means and how data is identified as abnormal.

-Page 3 Line 34: "Optimisation was not carried out in effective manner" is missing an explanation why it is not effective. The same can be found on Page 4 Line 16

-Page 4 Line 17: What is the difference in this context between optimal sensor placement and sampling design?

Methodology

-In general, the methodology section is not described in an understandable way and not outlined clearly, hence, it does not allow a reproduction by fellow scientists.

-Page 4 Line 25 to 29 would better fit in the introduction section.

-It is not clear if the paper deals with leak detection or leak localisation, since the literature review deals one time with detection and then switches to localisation and vice versa. For example, sentences in the methodology section like on Page 4 Line 29: "in order to detect the leak location" are confusing.

-Page 5 Line 2: What is a water distributer system?

-Page 5 Line 1-5: Why do the authors introduce a graph description of a water distribution system if it is not used later on? Additionally, a reference is missing to previous literature of how to describe a water distribution system as a mathematical graph. For sure, this is not the idea of the authors.

-Page 5 Line 5: Reference to EPANET is missing (Rossman 2000)

-Page 5 Line 8: How are leaks simulated in this paper with EPANET? Why did the authors use extended-period simulations, it seems there is not need for this.

-An important parameter is the leak's size, because this parameter has an effect on the size of the pressure drop and hence the detectability, but the leak size is not mentioned throughout the paper at all. In fact, while reading the paper, it is not clear if there are any leak simulations performed at all. If that is the case, the whole method proposed by the authors is very questionable, because the definition of normal and abnormal pressures does not make sense at all. Can the authors please clarify this point, because it is crucial for the whole publication?

-Figure 1: Besides the bad resolution and that some of the text in the figure is cut away, the figure is not very informative. What does "abnormal pressure data nodes are distributed" even mean?

[Figure]

-The enhanced brownboost classifier method is missing a crucial citation to the original paper by (Freund 2000), who invented this classification method. This is a clear case of plagiarism.This situation is further aggravated by the fact that this is one of the two key methods in this paper.

-It is not clear why the authors use a brownboost classifier at all, since it is invented for noisy environments. The authors are testing their method on simulations, which are not noisy. What is the reason why this classifier was chosen and no other one?

-It is not clear why a k-NN classifier is used before the brownboost classifier. Is it a k-NN classifier or is it just the application of equation 1 on the pressure data?

-It is not clear why the outcome of Equation 1 on Page 7 is binary (0 or 1 as stated on Page 7 Line 19). Looking at the equation, the outcome is supposed to be a floating point number between 0 and 1.

-Page 9 Line 10: How and to what extend does the brownboost classifier improve the classification accuracy and compared to what?

-Similarly to the brownboost classifier method, once again, the glowworm swarm optimisation model is missing a crucial citation to the original work by (Krishnanand and Ghose 2006), the inventors of this algorithm. All the equations in this section can be found in the paper of Krishnanand and Ghose 2006. This is the second clear case of plagiarism which is again aggravated by the fact that this is the second of two key methods in the author's work! Hence, both key methods of this paper are presented in such a way that they were developed by the authors, but in fact, they were not. In total, the paper contains 8 full pages (pages 6 to 13) copying ideas from other authors without citing them adequately.

-Page 11 Equation 11: Parameter gamma is not defined or mentioned in the text. Furthermore, maybe the most important part of a sensor placement algorithm is how to compute the objective function. It is not clear through the whole paper how the

authors actually compute the this function nor what the objective function means in the context of this paper at all.

-Page 12 Equation 13: There is an error in the equation. L_b(t) is in the subscript of the sum.

-Algorithm 2: Since glowworm swarm optimisation is a heuristic method, it cannot be guaranteed that it leads to the optimal solution / optimal node for sensor placement.

Simulation settings:

-In my opinion the machine on which the algorithm has been implemented is not important if the computation time is not discussed in the result section.

-The settings of the constants in the optimisation algorithm (beta, gamma, rho) is not mentioned here, but for optimisation this is of high interest.

-The paper is missing a figure showing the DMA in Barcelona crucial for a further understanding of the results of this paper. Furthermore, it is not clear how the authors get the hydraulic network. Did they get it from researchers in Barcelona? Then it might be also necessary to cite the publication where this network has been introduced for the first time.

-Figure 4: The resolution of the figure is very bad. This has to be improved. Additionally, the figure shows a standard EPANET network (Net 3). Looking at this figure and the fact that the Barcelona network is missing, it is not clear to the reviewer if the authors actually used the Barcelona network for the simulations in this paper, since important materials (Barcelona network model) is not shown.

-Page 14 Line 5-11: It is not clear why the authors have chosen the abbreviations, for example, RM for "node with demand"?

Simulation results:

-In my opinion the convergence and convergence speed of the optimisation method is

of interest, but not mentioned here.

-It is not clear why the authors have chosen the two methods (SVM and Graph-partitioning) for comparison of their method? It seems that these methods are chosen at random from literature? Why haven't the authors chosen other methods that might perform better?

-It is not clear how the authors decide between normal and abnormal pressure data? What does it even mean in this context? Pressure in WDS is also dependent, where in the system it is measured (elevation, roughness values of pieps, valve settings, . . .) so just classifying points according to their pressure won't result in finding leaks automatically. Did the authors generate data by simulating leaks? How many leaks where simulated? What was the leak size?

-In general, it is not clear throughout the paper how the results are generated. The paper shows only sample calculations without detailed explanation. After the sample calculations, tables are listed with numbers and it is not clear, where this numbers come from.

-Using classification time as a measure for the performance of the algorithms is in context of sensor placement very questionable. Furthermore, the reviewer does not see the benefit of a classification time being 36 ms in contrast to 72 ms, since both are very fast. The interesting question would be the convergence time of the optimal sensor placement method, which is not listed in this paper.

-Does the number of pressure data in Figures 5 to 8 correspond to the number of sensors? If that is the case the method would be useless, because deploying 50-500 pressure sensors in a water distribution system of total pipe length of 17 km results in a pressure sensor every 340 to 34 meter. This is a highly unrealistic number of sensors for such a small distribution system. Certainly, no water utility would be able to afford that number of sensors.

-For optimal sensor placement algorithms the most interesting outcome is the location where sensors should be placed. The optimal sensor positions are not shown in this paper.

-A final comment about the use of abbreviations: The authors define abbreviations like EBBC-GWO multiple times in the paper without using it. Basically, in each section the abbreviations are defined again, which is certainly not the purpose of abbreviations at all.

-Finally, there are a lot of repetitions of paragraphs, hence, the paper is not concise.

Freund, Y. (2001). An adaptive version of the boost by majority algorithm. Machine Learning, 43(3), 293–318. http://doi.org/10.1023/A:1010852229904

Krishnanand, K. N., & Ghose, D. (2006). Glowworm swarm based optimization algorithm for multimodal functions with collective robotics applications. Multiagent and Grid Systems, 2(3), 209–222. http://doi.org/10.3233/MGS-2006-2301

Rossman, L A. EPANET 2 USERS MANUAL. U.S. Environmental Protection Agency, Washington, D.C., EPA/600/R-00/057, 2000.

Puust, R., Kapelan, Z., Savic, D. A., & Koppel, T. (2010). A review of methods for leakage management in pipe networks. Urban Water Journal, 7(1), 25–45. http://doi.org/10.1080/15730621003610878

---

## Author Comment (AC1) · 17 Dec 2018

1.The abstract is too long, and should not focus too much on the description of the methodology

Corrections are addressed in abstract

2. Shorten introduction and make the description of the various methodologies more condensed at analyse the short-comings at once, so that it gives a good reason for the adopted approach.

[Figure]

Introduction is shortened and various methodologies are condensed in section 1.

3. Introduction and "related work" should be combined, because in the introduction related work should already be cited to set the boundaries for the proposed approach.

Introduction and related work should be combined in section 1.

4.Mention at the end of the introduction what is the reason to develop the proposed approach (and what is new).

Corrections are addressed in section 1.

5.Avoid in the methodology section using s "code" and too detailed descriptions in the main text (preferably in supplementary data/information), but concentrate more on how the calibration/validations are done.

Corrections are addressed throughout the paper.

6. Are real pressure data used or is it all simulated data, e.g.? –

Simulated data is used to implement the proposed work. Eg. EBBC-GWO method is used for District Metered Areas (DMA) in Barcelona water distribution network. Total extent of DMA is 17.4 km of pipelines with 883 nodes, 927 pipes which distributes water to 639 consumers. The pipe diameters varied from 70 to 400 mm. The pressure at night flow changes between 29.31m and 43.46 m. Flow and pressure are identified at inflow and outflow point. DMA includes 311 nodes with demand (RM), 60l nodes without demand (EC), 48 hydrant nodes without demand (HI type), 14 dummy valve nodes without demand (VT) and 448 dummy nodes without demand (XX type). The water pipeline has two inflow inputs (i.e., reservoir nodes). Leak detection is based on the damage (leaks) in different locations of piping network includes the liquid outflow at leak location that changes the flow features (pressure heads, flow rates, acoustics signals, etc.) at monitoring locations of piping network. It is imagined that leaks occur at XX type nodes in which 448 potential leaks are detected. Leaks occur at any node or pipe.

7. In the results section new definitions should be avoided (should be introduced in Methodology section).

Definitions are avoided in result section and it is introduced in section 2.2.1, 2.2.2.

8. Also avoid repetition and figures and tables with same information. It is suggested to present one large table with all data in it and then discuss the differences of the three methods at once.

Corrections are addressed in section 4.1, 4.2,4.3,4.4.

9. It is not clear from the results how the localization of the sensors is determined - It should be good to discuss the implementation of the proposed approach in practice.

Corrections are addressed in section 4.4.

10. Is it for example needed to run the EPANET model continuously to detect anomalies? –

Yes, EPANET model is continuously needed to run for detecting anomalies.

11. Check tenses: past tense when it is the author's work, present tense when it is general knowledge - Check abbreviations: introduce once and then always use abbreviation (not introducing again) – Check language in general.

Corrections are addressed throughout the paper.

Specific comments:

1.Page 1, Line 26, unclear what is meant

Corrections are addressed page 1, section 1.

2. Page 1, line 27 give references -

Corrections are addressed page 1, section 1.

3. Page 1, line 28 evade=avoid

Corrections are addressed in page 1.

4. Page 2, line 1 "..in (kang et al: : :.) that.." should be something like "Kang et al. (2018) found: : :" check rest of the document too.

Corrections are addressed in page 2 and throughout the paper

6.Page 2, line 33 introduce abbreviation –

Abbreviation is already introduced in page 1.

7. Page 3, line 5-18, is more for the methodology section

Corrections are addressed in section 1.

8. Page 4, line 26, give references (not always the case: : :) –

Corrections are addressed in section 2.

9. Page 9, line 4-5 this statement should be proven by the results (and is not adequate here).

Corrections are addressed in section 2.2.1

10. Page 14, Figure 4 is not readable –

Figure 4 is simulation diagram which is enlarged.

11. Page 15, line 2 Explain better (in methodology section) what is meant by "classified"

Corrections are addressed in section 2.2.1

12. Page 24, line 29-31 Not clear what is meant: : :

Corrections are addressed

13. Only less false positive or also better on other 3 performance indicators?

No, all the indicators provide better performance while detecting water leakage.

14. What is meant by "experimental evaluation"? What experiments are executed?

Corrections are addressed in section 5.

Please also note the supplement to this comment:
https://www.drink-water-eng-sci-discuss.net/dwes-2018-19/dwes-2018-19-AC1-supplement.pdf

———————————————

[Figure]

**Supplement:**

//Anonymous referee #1 corrections are addressed in red color

**Hybridisation of brownboost classifier and glow-worm swarm based optimal sensor placement for water leakage detection**

5 **Abstract.** Water Distribution System distributes the water to customer with the better quality and pressure. Due to the leakage, the sufficient amount of water is not delivered to the consumer. Many researchers introduced the techniques for detecting the water leakage in distribution system. But, the water leakage detection accuracy was not improved and time consumption was also not reduced. To improve the water leakage detection performance, Enhanced BrownBoost Classifier based Glowworm Swarm Optimization (EBBC-GWO) Method is introduced. Enhanced BrownBoost Classifier model 10 considers k-Nearest Neighbor (k-NN) classifier as weak classifier. It combines all k-NN classifier to construct strong classifier for classifying normal data or abnormal data with higher accuracy. After classification, optimization process is executed where every solution corresponds to the glowworm (i.e., abnormal pressure data node) in search space. Glowworm updates its location to the glowworm in dynamic decision space and optimal one is selected for water leakage detection. By this way, water leakage detection accuracy is improved with lesser false positive rate. Experimental results demonstrated that 15 EBBC-GWO method is higher in case of classification accuracy, false positive rate, classification time and water leakage detection accuracy.

**1 Introduction**

Water Management System (WMS) is efficient through the development of automated system for leakage detection in water network. When the leaks are detected early, corrective actions are carried out to avoid wastage of natural resources 20 and economical losses (Perfido et al. 2017). Leaks in Water Distribution Network (WDN) are detected through machine learning techniques. A fast water leakage detection system with adaptive design was introduced in (Kang et al. 2018) that combined one-dimensional convolutional neural network and support vector machine (1D-CNN-SVM) model. An actual water pipeline network was constructed in graph network and leakage occurred at virtual points on graph. However, the water leakage detection accuracy was not improved. A multi-stage graph partitioning approach was introduced in 25 (Rajeswaran et al. 2018) to find the off-line flow measurements were required for reducing cost. But, the classification performance was not carried out for water leak detection.

[revised manuscript text omitted]

The certain problems are identified from the existing methods are lesser classification accuracy, higher classification time consumption, lesser water leakage detection accuracy, higher false positive rate, and so on. In order to overcome these problems, proposed EBBC-GWO method is developed with the objective of increasing classification accuracy with minimum time while detecting water leakage.

- Enhanced BrownBoost Classifier model in EBBC-GWO takes k-NN classifier as weak and categorize training samples by majority vote of neighbor. Then the strong classifier is constructed by combining all weak classifiers. This in turns, classification accuracy is increased with minimum time.
- In Glowworm Swarm Optimization model is used to find the optimal node to place the sensor for detecting water leakage. This is done by computing objective function of each glowworm.

[revised manuscript text omitted]
 to select the optimal one with minimum false positive rate and maximum water leakage detection accuracy. False positive rate is measured as the leak which is incorrectly detected through the sensor placed nodes. Whereas, water leakage detection accuracy is the correct detection of leak by placing sensor at optimal nodes. The agents in GSO are the abnormal pressure data nodes. Every glowworm estimates their objective function and sends the information of current location to their neighbors based on objective function. Lesser objective function nodes get attracted by brighter objective function neighbors. Every glowworm takes the decision through probabilistic mechanism for moving to its neighbor with higher objective function. After that, the position of nodes gets updated and the optimal node is selected for the sensor placement to detect the leakage in WDS. By this way, the water leakage detection accuracy is improved.

**3 Simulation Settings**

[revised manuscript text omitted]

Table 4 describes the false positive rate of three different techniques for number of sensor placement nodes ranging from 50-500. The performance of false positive rate using EBBC-GWO Method is compared with the existing 1D-CNN-SVM model (Kang et al. 2018) and Multi-Stage Graph Partitioning Approach (Rajeswaran et al. 2018). From the table, it is clear that the false positive rate of EBBC-GWO Method is lesser than other existing techniques (Kang et al. 2018), (Rajeswaran et al. 2018).

The localization of sensor is done by identifying optimal nodes in WDN. At first, pressure data is collected from the nodes. Then the classification process is carried out to classify the data as the normal data or abnormal data for performing the optimization process. An optimization process is carried out to identify the data as optimal one. An enhanced brownboost classifier is classifies the data and glowworm swarm optimization optimizes the data to detect the water leakage in our research work. Enhanced brownboost classifier combines all kNN classifiers results to form the strong classifier output. Then, the abnormal pressure data is distributed and taken as glowworms. Besides, objective function (luciferin value) for each glowworm is computed where lesser brighter glowworm attracted by higher brighter glowworm and then updates its position. This helps to identify the optimal node for sensor localization. Finally, water leakage gets detected by placing the sensor at that particular node. By this way, the false positive rate gets reduced in our research work.

[revised manuscript text omitted]

25  Tianjin. Univ., 17, 181 - 186, doi: 10.1007/s12209-011-1594-4, 2011.

30

---

## Author Comment (AC2) · 19 Dec 2018

General Comments:

1. The results are recommending to put 50 to 500 sensors in a network of 17 km (up to a sensor every 34 meter) are without doubt unfeasible for applying the methodology on real-world systems.

EBBC-GWO method used Barcelona water distribution network. From these source 4645 km of pipeline is considered. It consists of 883 nodes, 927 pipes which distributes

water to 639 consumers. [Addressed in section 3]

Specific Comments:

Introduction And Related Work

2. Page 1 Line 28: The sentence "leaks in WDN are detected through machine learning techniques" is not true, these methods are just one way to detect leaks, in fact, used only in scientific literature and not in practice at all.

In our work, water leakage detection is performed using machine learning techniques. Therefore, 'leaks in WDN are detected through machine learning techniques' is included in introduction. As per your requirement, the introduction is shortened. Therefore, the above said sentence is ignored in introduction.

3. In general, the listed literature is a mix of different methods for leak detection, which are even focusing on different physical effects caused by a leak (e.g. flow, noise, pressure without even mentioning it here) as well as a wild mix between leak localisation, leak detection and optimal sensor placement methods. Therefore, they can't be compared and listed in the way they are presented in introduction and the related works section .

Corrections are addressed in section 1

4. The literature review is insufficient: Only very recent literature is listed . The oldest publication is from 2015. Older key literature as well as novel key literature for this topic is missing. In my opinion, the state of the art is not well described. I recommend the authors to review the literature in the papers that the mention and identify the key papers for this topic.

The state of the art methods are well defined in section 1. older papers are added in section 1.

5. Page 3 Line 8: Throughout the paper it is mentioned that the method allows to detect

leaks with higher accuracy, but it is not clear what is meant. Higher than what?

Higher accuracy means higher classification accuracy. The proposed method classifies the data as normal or abnormal pressure data with classification accuracy (i.e., 20% higher than existing methods ).

6. Page 3 Line 23: Related works: Is the work listed in the introduction not also related work? What is the difference of this section to the previous one?

The work listed in the introduction also the related work. The work listed in the introduction is more related to our work. Therefore, they are listed in the introduction and the remaining paper is listed in the related work section.

As per the comment from anonymous referee #1 the introduction and related work sections are combined.

7. -Page 3 Line 13: The paper states the difference between normal data and abnormal data, but it is not clear throughout the paper what this terms actually means and how data is identified as abnormal.

The pressure flow is normal (i.e., maintained by a constant pressure) in the collection pressure data it is classified or identified as normal pressure data. Whereas, the pressure flow is abnormal (not a constant value) in the collection of pressure data is identified as abnormal pressure data. The abnormal pressure occurs when the pipe break, water leakage or fault in the pipeline is identified.

8. Page 3 Line 34: "Optimisation was not carried out in effective manner" is missing an explanation why it is not effective. The same can be found on Page 4 Line 16

But, the optimization was not carried out in effective manner because it failed to consider the real-time operating pressure and flow data in designed approach. [ it is addressed in page 3 line 5-6].

The same can be found on Page 4 Line 16

But, the performance of leak identification was not carried out in effective manner because it varied based on the timing and duration of the measurement.[ it is addressed in page 3 line 22-23].

9. Page 4 Line 17: What is the difference in this context between optimal sensor placement and sampling design?

Sampling Design

Sampling design (SD) used to perform localization and quantification of pressure sensors in WDS for leak detection. SD was derived based on the criteria of maximization of total leak sensitivity, sensitivity consistence, minimization of information redundancy and sensors number criteria. However, the classification time was not reduced by using SD method.

Optimal Sensor Placement

Optimal sensor placement is also used to perform localization for leak detection. Here, sensor placement problem was formulated as an integer optimization problem. The optimization criterion was based on minimizing the number of non-isolable leaks according to the isolability criteria. However, sensor placement approach was not effective in detecting the water leak Methodology

10. It is not clear if the paper deals with leak detection or leak localisation, since the literature review deals one time with detection and then switches to localisation and vice versa. For example, sentences in the methodology section like on Page 4 Line 29:"in order to detect the leak location" are confusing

As per your requirement, in order to detect the leak location sentence is changed as In order to perform the leak detection, EBBC-GWO Method is introduced. [it is addressed in page 4 line 17]. The objective of the research work is to perform optimal sensor placement for water leakage detection. Therefore, the literature review deals optimal sensor placement/localization and leak detection concepts.

11. Page 5 Line 2: What is a water distributer system?

Typing error water distribution system [correction addressed in page 4 line 28]

12. Page 5 Line 1-5: Why do the authors introduce a graph description of a water distribution system if it is not used later on? Additionally, a reference is missing to previous literature of how to describe a water distribution system as a mathematical graph. For sure, this is not the idea of the authors.

A water distribution system comprises the set of graph branches (i.e.,G=(V,E)) where E denotes set of edges (i.e., pipes) and V denotes the set of vertices (i.e., nodes) explaining about the pipe connections and endings in (R. Sarrate et al. 2014). A water distribution system is a collection of pipes (i.e., links) connected to the nodes (junctions, tanks and reservoirs). The water flows are computed through hydraulic balancing, and through solving the equations at every node and links. EPANET software works out the network hydraulic equations automatically (Rossman 2000). By using the graph model, EPANET software obtains the pressure and flow rate of every node and links from hydraulic simulation. [Corrections is addressed in page 5]

13. Page 5 Line 5: Reference to EPANET is missing (Rossman 2000)

Reference included in page 4 line 32

14. Page 5 Line 8: How are leaks simulated in this paper with EPANET? Why did the authors use extended-period simulations, it seems there is not need for this.

EPANET software tool is simulated at without leak condition and pressure data readings are collected. During the simulation, it detects the flow of water in pipe, the pressure at each node and height of the water in container

15. An important parameter is the leak's size, because this parameter has an effect on the size of the pressure drop and hence the detectability, but the leak size is not mentioned throughout the paper at all. In fact, while reading the paper, it is not clear if there are any leak simulations performed at all. If that is the case, the whole method

proposed by the authors is very questionable, because the definition of normal and abnormal pressures does not make sense at all. Can the authors please clarify this point, because it is crucial for the whole publication?

Leak's size parameter is included in section 4.5. The size of the leakage was 142meter, 0.99meter, 0.99meter, 12.31meter. [addressed in section 3]

The pressure flow is normal (i.e., maintained by a constant pressure) in the collection pressure data it is classified or identified as normal pressure data. Whereas, the pressure flow is abnormal (not a constant value) in the collection of pressure data is identified as abnormal pressure data. The abnormal pressure occurs when the pipe break, water leakage or fault in the pipeline is identified. [Corrections is addressed in page 6].

16. Figure 1: Besides the bad resolution and that some of the text in the figure is cut away, the figure is not very informative. What does "abnormal pressure data nodes are distributed" even mean?

Figure corrected in page 5.

17. It is not clear why the authors use a brownboost classifier at all, since it is invented for noisy environments. The authors are testing their method on simulations, which are not noisy. What is the reason why this classifier was chosen and no other one?

Brownboost classifier is a robust boosting and higher accurate classifier. It boost all weak (i.e., base) classifiers and combine to make strong one. Besides, the time parameter and error rate is highly concentrated in brownboost classifier. This in turns, the performance of classification is improved with minimum. Therefore, Enhanced Brownboost classifier is used in EBBC-GWO method for classifying the collection of pressure data

18. It is not clear why a k-NN classifier is used before the brownboost classifier. Is it a k-NN classifier or is it just the application of equation 1 on the pressure data?

The brownboost classifier performed to boost all the base classifiers., In EBBC-GWO, k-NN classifier is considered as the base classifier since the data in the network are not linearly separable. Therefore, k-NN classifier is used before the brownboost classifier

19. It is not clear why the outcome of Equation 1 on Page 7 is binary (0 or 1 as stated on Page 7 Line 19). Looking at the equation, the outcome is supposed to be a floating point number between 0 and 1.

In proposed work, the collection of data is classified as either normal or abnormal. The floating point value is rounded off in proposed work (i.e., 0.4 is 0 and 0.6 is 1).

20. Page 9 Line 10: How and to what extend does the brownboost classifier improve the classification accuracy and compared to what?

Brownboost classifier includes the ability to learn a collection of pressure data into a fixed level of accuracy thus classifies the data with classification accuracy than the state-of-the-art methods. [Correction addressed in page 9 line 12] Brownboost classifier is able to convert weak classifier into strong classifier. In addition voting process is carried out to classify the pressure data of node thus increases the classification accuracy in brownboost classifier

21. The enhanced brownboost classifier method is missing a crucial citation to the original paper by (Freund 2000), who invented this classification method. This is a clear case of plagiarism. This situation is further aggravated by the fact that this is one of the two key methods in this paper

Reference Included in page 6 line 20

22. Similarly to the brownboost classifier method, once again, the glowworm swarm optimization model is missing a crucial citation to the original work by (Krishnanand andGhose 2006), the inventors of this algorithm.

Reference Included in page 10 line 2

23. Page 11 Equation 11: Parameter gamma is not defined or mentioned in the text. Furthermore, maybe the most important part of a sensor placement algorithm is how to compute the objective function. It is not clear through the whole paper how the authors actually compute this function nor what the objective function means in the context of this paper at all.

Parameter gamma is not defined or mentioned in the text. [Correction addressed in page 12]

The objective function calculation is addressed in equation (12)

24. Page 12 Equation 13: There is an error in the equation. L_b(t) is in the subscript of the sum.

Correction addressed in equation 14.

25. Algorithm 2: Since glowworm swarm optimisation is a heuristic method, it cannot be guaranteed that it leads to the optimal solution / optimal node for sensor placement

Glowworm swarm optimization performed to provide the best corrective measures to designers when assigning correct heuristic. In proposed Glowworm swarm optimization, the objective function calculation is used to provide value of identifying optimal sensor placement. Moreover, it is cheap, simple and fast method to provide better solution for optimal sensor placement.

Simulation settings:

26. The settings of the constants in the optimisation algorithm (beta, gamma, rho) is not mentioned here, but for optimisation this is of high interest.

Correction addressed in table 1

27. The paper is missing a figure showing the DMA in Barcelona crucial for a further understanding of the results of this paper. Furthermore, it is not clear how the authors get the hydraulic network. Did they get it from researchers in Barcelona? Then it might

be also necessary to cite the publication where this network has been introduced for the first time

Fig. 1 : Map of the portion of the Barcelona WDN Fig. 2: Simulation model of a portion of Barcelona WDN (arino et al.2017) Ramon Ariño, Jordi Meseguer, Ramon Pérez and Joseba Quevedo, "Case Studies", Springer, Real-time monitoring and operational control of drinking water systems,

28. Page 14 Line 5-11: It is not clear why the authors have chosen the abbreviations, for example, RM for "node with demand"? Flow and pressure are identified at inflow and outflow point. DMA includes 311 nodes with demand (RM type), 60l nodes without demand (EC type), 48 hydrant nodes without demand (HI type), 14 dummy valve nodes without demand (VT type) and 448 dummy nodes without demand (XX type) (R. Sarrate et al. 2014). [Correction addressed in section 3] RM for "node with demand is referred from (R. Sarrate et al. 2014).

29. Figure 4: The resolution of the figure is very bad. This has to be improved. Additionally,the figure shows a standard EPANET network (Net 3). Looking at this figure and the fact that the Barcelona network is missing, it is not clear to the reviewer if the authors actually used the Barcelona network for the simulations in this paper, since important materials (Barcelona network model) is not shown

quality of improved Figure 4 in section 3. In our work we consider District Metered Areas (DMA) in Barcelona network.

From the above Fig. 3: The small area that contains 100 sensor nodes are considered to perform simulation in figure 4 a and b Simulation results:

30. -In my opinion the convergence and convergence speed of the optimization method is of interest, but not mentioned here.Convergence and convergence speed of EBBC-GWO

[Addressed in section 4.3] Fig. 4:

31. It is not clear how the authors decide between normal and abnormal pressure data? What does it even mean in this context? Pressure in WDS is also dependent, where in the system it is measured (elevation, roughness values of pieps, valve settings,: : :) so just classifying points according to their pressure won't result in finding leaks automatically. Did the authors generate data by simulating leaks? How many leaks where simulated? What was the leak size?

The pressure flow is normal (i.e., maintained by a constant pressure) in the collection pressure data it is classified or identified as normal pressure data. Whereas, the pressure flow is abnormal (not a constant value) in the collection of pressure data is identified as abnormal pressure data. The abnormal pressure occurs when the pipe break, water leakage or fault in the pipeline is identified. [Correction is addressed in page 6].

Pressure in WDS is also dependent where in the system it is measured, so just classifying points according to their pressure. By considering the classification result, the pressure data is identified as normal or abnormal. From that the leak is detected. The authors generate data by simulating leaks. Five leaks were simulated. The size of the leakage was 142meter, 0.99meter, 0.99meter, 12.31meter. [addressed in section 3]

32. It is not clear why the authors have chosen the two methods (SVM and Graph-partitioning) for comparison of their method? It seems that these methods are chosen at random from literature? Why haven't the authors chosen other methods that might perform better?

Based on the objective of the proposed EBBC-GWO framework (i.e., increase the classification accuracy of water leakage detection with minimum time), the existing methods such as 1D-CNN-SVM model and Multi-Stage Graph Partitioning Approach are taken as base paper. The proposed work concept is derived by considering problems of these base papers. The drawbacks of these methods are effectively convinced by implementing proposed work. These two base papers are explained to understand the

proposed work and these are more related to the proposed work. Therefore, 1D-CNN-SVM model and Multi-Stage Graph Partitioning Approach are chosen for comparison purpose.

33. -In general, it is not clear throughout the paper how the results are generated. The paper shows only sample calculations without detailed explanation. After the sample calculations, tables are listed with numbers and it is not clear, where this numbers come from.

The results are generated form MATLAB simulations and these results (i.e., numbers) are tabled in the paper. Sample calculation with detailed explanations is given in section 4.1.

34. -Using classification time as a measure for the performance of the algorithms is in context of sensor placement very questionable. Furthermore, the reviewer does not see the benefit of a classification time being 36 ms in contrast to 72 ms, since both are very fast. The interesting question would be the convergence time of the optimal sensor placement method, which is not listed in this paper.

The convergence time of optimal sensor placement method using EBBC-GWO Method is 1.3ms [addressed in section 4.2]

35. Does the number of pressure data in Figures 5 to 8 correspond to the number of sensors? If that is the case the method would be useless, because deploying 50-500pressure sensors in a water distribution system of total pipe length of 17 km results in a pressure sensor every 340 to 34 meter. This is a highly unrealistic number of sensors for such a small distribution system. Certainly, no water utility would be able to afford that number of sensors.

The number of pressure data in Figures 5 to 8 is not correspond to the number of sensors. The one sensor node sense the number of pressure data. EBBC-GWO method used Barcelona water distribution network. From these source 4645 km of

pipeline is considered. It consists of 883 nodes, 927 pipes which distributes water to 639 consumers. [Addressed in section 3]

36. For optimal sensor placement algorithms the most interesting outcome is the location where sensors should be placed. The optimal sensor positions are not shown in this paper.?

Glowworm swarm optimization calculates the objective function for neighboring node and current node. If the value of objective function is higher than the current node, the neighboring node is considered as optimal node to place the sensor for leakage detection. Fig. 5:

As shown Fig. 5: red color denotes the optimal sensor placement. [addressed in section 3]

37. A final comment about the use of abbreviations: The authors define abbreviations like EBBC-GWO multiple times in the paper without using it. Basically, in each section the abbreviations are defined again, which is certainly not the purpose of abbreviations at all.

Corrections are addressed throughput the paper

38. Finally, there are a lot of repetitions of paragraphs, hence, the paper is not concise

Corrections are addressed throughout the paper.

Please also note the supplement to this comment:
https://www.drink-water-eng-sci-discuss.net/dwes-2018-19/dwes-2018-19-AC2-supplement.pdf

[Figure]

[Figure]

Fig. 1. Map of the portion of the Barcelona WDN

**Fig. 2.** Simulation model of a portion of Barcelona WDN (arino et al.2017)

[Figure]

**Fig. 3.** Barcelona water network

[Figure]

**Fig. 4.** Objective function vs Iteration Count

**Fig. 5.** optimal sensor placement in Water Distribution Network

**Supplement:**

//Anonymous Referee #2  corrections are addressed in blue color

**Hybridisation of brownboost classifier and glow-worm swarm based optimal sensor placement for water leakage detection**

Rejeesh Rayaroth [1] , Sivaradje Gopalakrishnan[1]

5 Department of Electronics & communication Engineering, Pondicherry Engineering College, Puducherry, 605014, India

**Abstract.** Water Distribution System distributes the water to customer with the better quality and pressure. Due to the leakage, the sufficient amount of water is not delivered to the consumer. Many researchers introduced the techniques for detecting the water leakage in distribution system. But, the water leakage detection accuracy was not improved and time 10 consumption was also not reduced.  To improve the water leakage detection performance, Enhanced BrownBoost Classifier based Glowworm Swarm Optimization (EBBC-GWO) Method is introduced. Enhanced BrownBoost Classifier model considers k-Nearest Neighbor (k-NN) classifier as weak classifier. It combines all k-NN classifier to construct strong classifier for classifying normal data or abnormal data with higher classification accuracy. After classification, optimization process is executed where every solution corresponds to the glowworm (i.e., abnormal pressure data node) in search space. 15 Glowworm updates its location to the glowworm in dynamic decision space and optimal one is selected for water leakage detection. By this way, water leakage detection accuracy is improved with lesser false positive rate. Experimental results demonstrated that EBBC-GWO method is higher in case of classification accuracy, false positive rate, classification time and water leakage detection accuracy than the state-of-the-art method.

**1 Introduction**

20 Water Management System is efficient through the development of automated system for leakage detection in water network. Water leaks in Water Distribution Network (WDN) leads to economic losses for final consumer in huge quantities of fluid transportation. Therefore, finding leaks is important one in water network. In different WDN, losses because of leaks are calculated to 30% of extracted water. Water distribution is installed with underground pipes. The underground pipeline monitoring is difficult than open space water pipelines. Leaks in pipes are due to many several factors like pipe's age, 25 improper fitting and disasters (Sadeghioon et al. 2018). Leak management in pipe networks are developed in (Puust et al. 2010) through leakage assessment methods. A solution is needed to identify and to determine leak in the water network. When the leaks are detected early, corrective actions are carried out to avoid wastage of natural resources and economical losses (Perfido et al. 2017).

A fast water leakage detection system with adaptive design was introduced in (Kang et al. 2018) that combined one-30 dimensional convolutional neural network and support vector machine (1D-CNN-SVM) model. An actual water pipeline network was constructed in graph network and leakage occurred at virtual points on graph. However, the water leakage

detection accuracy was not improved. In order to improve the water detection accuracy, a multi-stage graph partitioning approach was introduced in (Rajeswaran et al. 2018) to find the off-line flow measurements were required for reducing cost. But, the classification performance was not carried out for water leak detection.

To improve the classification performance, a new statistical framework was introduced in (Fagiani et al. 2016) for leakage identification. The framework comprises three sections for extraction and selection of features at leakage detection. But, the classification accuracy was not improved using new statistical framework. Leakage detection and localization on water transportation was introduced in (Kayaalp et al. 2017) to recognize and find leaks on water pipelines positions with pressure data. Three multi-label classification techniques like RA-kELd, BRkNN, and BR with SVM were employed for leak detection. The false positive rate was not minimized in this system. Therefore, a new leak detection technique was presented in (Sadeghioon et al. 2018) for water distribution pipelines as it buried pressurized fluid flow pipe. The detection technique was dependent on pressure sensor connected with temperature difference. The pipe failure detection accuracy was improved with no false positives. But, the time taken for water leakage detection was not reduced using anomaly detection algorithm.

To solve the above issue, nominal variable was developed in (Dawidowicz 2017) with artificial neural network. The developed scheme also addresses the issues with pressure and division of water distribution system into the pressure zones. The classification was carried out depending on neural network variables explaining particular parameters that change the water distribution system design. However, the calculation of accuracy was not efficient. To address this issue, a new model was introduced in (Hajibandeh 2018) for leakage amount and location detection. A nodal pressures and demands were adjusted through multi-objective ant colony based optimization model. But, leakage detection time consumption was not reduced using ant colony based optimization model.

To analyze the time consumption, an algorithm structure was constructed in (Kumar et al. 2017) with modularity of wavelet and neural network that join the capability of wavelet transform through examining leakage signals and classification ability of artificial neural networks. The study authenticated that time domain was not apparent to features concerning noisy leak signals and significance of selection of mother wavelet in water distribution pipes. Yet another time consumption issue was addressed by Leak detection technique in (Martini et al. 2015) through vibration monitoring methods. The long-term objective was attained for automatic detection of leaks. The leak detection accuracy was not improved for automatic detection. To overcome this limitation, the designed approach was introduced in (Sousa et al. 2015) depending on the steady-state modeling through monitoring tank flow and pressure at strategic nodes. The selection of pressure monitoring nodes was performed with graph theory ideas in WDN. But, the classification process was not carried out in designed approach.

To address the classification process, a new method was introduced in (Seyoum et al. 2017) for household leakage detection through sound signal recordings. The designed approach comprised recording of sound signals created through water fixtures and appliances. The recordings were employed to identify abnormal situation (i.e., leak). But, the classification accuracy was not improved. To solve the above said problem, a new methodology was introduced in

(Steffelbauer and Fuchs-Hanusch 2016) with uncertainties of different sources in optimal sensor placement issues for leak localization on potential pressure measurement points. The issues were addressed for diverse sensors and uncertainties. Yet another sensor placement optimization issues were addressed in (Christodoulou et al. 2013) in urban water distribution networks through entropy-based approach for feasible waterloss incident detection. The designed method was employed for

5    longitudinal for acoustic, pressure or flow sensors on pipe segments. But, the optimization was not carried out in effective manner because it failed to consider the real-time operating pressure and flow data.

To solve the above issue, a new sensor placement approach was introduced in (Casillas et al. 2013) for leak location in Water Distribution Networks (WDNs). The sensor placement issues were addressed through solving integer optimization problem. However, sensor placement approach was not effective in detecting the water leak. Therefore, Water leakage

10    detection techniques were introduced and issues were highlighted in (Adedeji et al. 2017). It accomplished the effort and development in leakage detection technologies. It was efficient detection of background kind leakages. But, time consumption was not reduced using water leakage detection techniques. To address this problem, a leakage detection method was introduced in (Pérez et al. 2009) depending on discrepancies among pressure measurements and estimations attained from calibrated water distribution network. Every sensor in network identified discrepancy in pressure because of leakage

15    based on its location.

Yet another leak detection method was introduced in (Zhang and Wang 2011) depending on Bayesian theory and Fisher's law for water distribution systems. A hydraulic model was linked with parameters of leaks. The randomness of parameter values was computed through the probability density function and Bayesian theory. The false positive rate was not lessened using leak detection method. In order to reduce the false positive rate, a factorized distribution assigned the

20    inflow demand across the consumption nodes with individual billing data and transmitted the demand across consumption nodes in (Moors et al. 2018). The automatic leak localization method with demand distribution models was developed to compute the leak localization performance. But, the performance of leak identification was not carried out in effective manner because it varied based on the timing and duration of the measurement. To solve this, the location of leak was determined in (Wachla et al. 2015) by group of neuro-fuzzy classifiers. The main aim of the method was to increase the

25    accuracy of leak detection and location. The designed classifier only predicts the leak in the one subarea of water distribution network. In order to overcome this problem, a novel method called robust leakage detection was introduced in (Kim et al. 2015) using pressure data. However, the leakage prediction with minimum time was not ensured.

To solve the above issue, A Sampling Design (SD) method was introduced in (Gamboa-Medina and Reis 2017) for localization and quantification of pressure sensors in WDS for leak detection. Sampling design is the selection of test

30    locations in water distribution system for data collection. SD addressed four criteria, namely improvement of leak sensitivity and sensitivity constancy and reduction of information redundancy and sensor count. Though, the iterative increase in the number of sensing devices/locations on a network, the SD was unable to be performed well. However, the classification time was not reduced by using SD method. To address this issue, a new technique was designed in (Islam et al. 2011) to identify and to diagnose the leakage in WDS. A fuzzy-based algorithm incorporated many uncertainties into many WDS parameters

like roughness, nodal demands and water reservoir levels. But, the water leakage detection accuracy was not improved by fuzzy-based algorithm.

The certain problems are identified from the existing methods are lesser classification accuracy, higher classification time consumption, lesser water leakage detection accuracy, higher false positive rate, and so on. In order to overcome these problems, proposed EBBC-GWO method is developed with the objective of increasing classification accuracy with minimum time while detecting water leakage.

- Enhanced BrownBoost Classifier model in EBBC-GWO takes k-NN classifier as weak and categorize training samples by majority vote of neighbor. Then the strong classifier is constructed by combining all weak classifiers. This in turns, classification accuracy is increased (i.e., 20% than the state-of-the-art-method) with minimum time.

- In Glowworm Swarm Optimization model is used to find the optimal node to place the sensor for detecting water leakage. This is done by computing objective function of each glowworm.

The paper is structured as follows; Section 2 explains EBBC-GWO Method. Section 3 discusses the simulation setting. Section 4 presents result analysis of proposed EBBC-GWO method with various parameters. In section 5, the conclusion of the work is given.

**2 Methodology**

In order to perform the leak detection, EBBC-GWO Method is introduced. The major contribution of the paper is described as follows. EBBC-GWO method comprises two models namely, Enhanced BrownBoost Classifier model and Glowworm Swarm Optimization model. Enhanced BrownBoost Classifier model uses k-Nearest Neighbor classifier as a weak. k-NN classifier is a non-parametric method to classify training samples by majority vote of neighbor with object being assigned to the class. After that, Brownboost combines the weak learner to form the strong classifier. By this way, the data are classified as normal data or abnormal data with higher classification accuracy. In Glowworm Swarm Optimization model, every solution corresponds to the glowworm in the search space. Every glowworm has the objective function for solving the optimization problem. Every glowworm sets the objective function at current location into luciferin value. Each glowworm functions in probabilistic manner to find the neighbor with higher luciferin value and travels toward it. Glowworm updates its location to the glowworm in dynamic decision space and optimal one is selected for water leakage detection.

**2.1 System Model**

[revised manuscript text omitted]

$$L_a(t + 1) = (1 - \rho)L_a(t) + \gamma J_a(t + 1) \tag{11}$$

From (11), $\rho$ denotes the luciferin decay constant. $\gamma$ symbolizes luciferin enhancement constant. $L_a(t)$ symbolizes luciferin value. $J_a(t)$ represent the objective function value at glowworm $a^{th}$ location at time t. From that, the objective function of water flow rate is calculated as follows,

$$O(f) = \frac{Velocity}{Time} \tag{12}$$

From (12), the objective function (i.e., higher flow rate) of each glow warm is calculated.

**Movement-phase**

[revised manuscript text omitted]

The simulation settings of the constants in the optimization algorithm such as beta, gamma, rho values are fixed in all simulations is shown in Table 1.

**Table 1 Values of parameters that are kept fixed in all the simulations**

| Beta '$\beta$' | Gamma '$\gamma$' | Rho '$\rho$' |
| --- | --- | --- |
| 0.08 | 0.6 | 0.4 |

4(a)

[Figure]

4(b)

**Figure 4 (a) and (b):Simulation Diagram of Water Distribution Network**

[Figure]

Figure 5 Simulation Diagram of optimal sensor placement in Water Distribution Network

**4 Simulation Result Analysis**

EBBC-GWO Method is designed for detecting the water leakage in WDS and compared with existing one-dimensional convolutional neural network and support vector machine (1D-CNN-SVM) model (Kang et al. 2018)  and Multi-Stage Graph Partitioning Approach (Rajeswaran et al. 2018). The efficiency of EBBC-GWO method is evaluated along with the metrics such as classification accuracy, classification time, water leakage detection accuracy and false positive rate.

**4.1 Classification Accuracy (CA)**

Classification accuracy is the one of the important parameter to evaluate the performance of proposed work.  It is measured in terms of percentage (%).  Classification accuracy is mathematically formulated by,

$$CA = \frac{Number\ of\ pressure\ data\ correctly\ classifed}{Total\ number\ of\ pressure\ data} * 100 \qquad (17)$$

**Sample calculation:**

➤ **1D-CNN-SVM model:** the number of pressure data correctly classified is 30 from simulation and the total number of pressure data is 50.  After that, classification accuracy is computed as,

$$CA = \frac{30}{50} * 100 = 60\ \%$$

➤ **Multi-Stage Graph Partitioning:** the number of pressure data correctly classified is 39 using simulation and the total number of pressure data is 50.  After that classification accuracy is calculated as,

$$CA = \frac{39}{50} * 100 = 78\ \%$$

➤ **Proposed EBBC-GWO Method:** the number of pressure data correctly classified is 45 using simulation and the total number of pressure data is 50. After that, the classification accuracy is obtained as,

$$CA = \frac{45}{50} * 100 = 90\ \%$$

When the classification accuracy is higher, method is efficient.

**Table 2: Tabulation for Classification Accuracy**

| Number of Pressure data (Number) | Classification Accuracy (%) | | |
|---|---|---|---|
| | 1D-CNN-SVM model | Multi-Stage Graph Partitioning Approach | EBBC-GWO Method |
| 50 | 60 | 78 | 90 |
| 100 | 58 | 80 | 96 |
| 150 | 67 | 82 | 90 |
| 200 | 68 | 82 | 93 |

| 250 | 80 | 90 | 98 |
| --- | --- | --- | --- |
| 300 | 80 | 92 | 96 |
| 350 | 80 | 85 | 94 |
| 400 | 85 | 91 | 98 |
| 450 | 83 | 89 | 96 |
| 500 | 88 | 93 | 98 |

Table 2 describes classification accuracy of three different methods implemented in Matlab simulation by taking diverse number of pressure data ranging from 50-500. The values in the table 1 are obtained from the results of simulation using EPANET software. The simulation result of classification accuracy using EBBC-GWO Method is compared with existing 1D-CNN-SVM model (Kang et al. 2018) and Multi-Stage Graph Partitioning Approach (Rajeswaran et al. 2018). The pressure data is taken from DMA in Barcelona water distribution network. In order to analyze the classification performance, the pressure data is gradually increased. While considering number of pressure data is 500, the classification accuracy is improved up to 88%, 93% and 98% using 1D-CNN-SVM model, Multi-Stage Graph Partitioning Approach and EBBC-GWO Method respectively. From the table, it clear that the classification accuracy using EBBC-GWO Method is higher than other existing techniques (Kang et al. 2018), (Rajeswaran et al. 2018).

[revised manuscript text omitted]

clear that the false positive rate of EBBC-GWO Method is lesser than other existing techniques (Kang et al. 2018), (Rajeswaran et al. 2018). Theses value are calculated from the results of simulation.

The localization of sensor is done by identifying optimal nodes in WDN. At first, pressure data is collected from the nodes. Then the classification process is carried out to classify the data as the normal data or abnormal data for performing the optimization process. An optimization process is carried out to identify the data as optimal one. An enhanced brownboost classifier is classifies the data and glowworm swarm optimization optimizes the data to detect the water leakage in our research work. Enhanced brownboost classifier combines all kNN classifiers results to form the strong classifier output. Then, the abnormal pressure data is distributed and taken as glowworms. Besides, objective function (luciferin value) for each glowworm is computed where lesser brighter glowworm attracted by higher brighter glowworm and then updates its position. This helps to identify the optimal node for sensor localization. Finally, water leakage gets detected by placing the sensor at that particular node. By this way, the false positive rate gets reduced in our research work.

The key aim of EBBC-GWO method is to place the minimal number of sensors at optimal nodes for detecting the leakages.

For conducting the simulation experiment, number of instances taken are 10. For every instance, the number of sensor placement nodes gets changed. When number of sensor placement data is taken as 350, EBBC-GWO Method attains the 3% of false positive rate while 1D-CNN-SVM model (Kang et al. 2018) and Multi-Stage Graph Partitioning Approach (Rajeswaran et al. 2018) produces 18% and 11% of false positive rate respectively. The false positive rate of EBBC-GWO Method is reduced by 69% and 54% compared to existing 1D-CNN-SVM model (Kang et al. 2018) and Multi-Stage Graph Partitioning Approach (Rajeswaran et al. 2018) respectively.

**4.5 Leakage size**

Lastly, leakage size is computed for analyzing the performance of water leakage detection. Leakage size is measured the amount water wasted through the time. Here, time is considered as 1 second. Leakage size is measured in terms of liters per second (lps).

$$Leakage\ size = \frac{amount\ of\ water\ wasted\ in\ the\ network}{Time} \qquad (21)$$

**Sample calculation:**

➢ **1D-CNN-SVM model:** the amount of water leakage is 15 lps and the time is considered as 1 sec. After that, leakage size is calculated as,

$$Leakage\ size = \frac{15}{1} = 15\text{lps}$$

➢ **Multi-Stage Graph Partitioning:** the amount of water leakage is 13 lps and the time is considered as 1 sec. After that, leakage size is calculated as,

$$Leakage\ size = \frac{13}{1} = 13\text{lps}$$

- **Proposed EBBC-GWO Method:** the amount of water leakage is 11 lps and the time is considered as 1 sec. After that, leakage size is calculated as,

$$Leakage\ size = \frac{11}{1} = 11\text{lps}$$

**Table 6: Tabulation for Leakage size**

| Number of Pressure Data (Number) | Leakage size (lps) | | |
|---|---|---|---|
| | 1D-CNN-SVM model | Multi-Stage Graph Partitioning Approach | EBBC-GWO Method |
| 50 | 15 | 13 | 11 |
| 100 | 17 | 15 | 13 |
| 150 | 20 | 16 | 14 |
| 200 | 22 | 20 | 18 |
| 250 | 23 | 18 | 16 |
| 300 | 22 | 21 | 19 |
| 350 | 24 | 22 | 18 |
| 400 | 24 | 23 | 20 |
| 450 | 23 | 22 | 21 |
| 500 | 25 | 24 | 22 |

Table 6 illustrates the leakage size of proposed and existing methods with respect to different number of pressure data. The number of pressure data is varied from 50-500 for performing simulation. The performance of leakage size is using proposed EBBC-GWO Method is compared with the existing 1D-CNN-SVM model (Kang et al. 2018) and Multi-Stage Graph Partitioning Approach (Rajeswaran et al. 2018). From the comparison, it is clear that the leakage size of EBBC-GWO Method is minimum than state-of-the-art methods (Kang et al. 2018), (Rajeswaran et al. 2018).

Enhanced BrownBoost Classifier model is employed to classify the pressure data is normal or abnormal through considering the k-Nearest Neighbor classifier as base classifier. This in turns, brownboost combines the weak learner to make the strong classifier. After that, optimization model is employed to find the optimal node for finding the sensor placement. From that, the leak in the water network is identified early using EBBC-GWO Method. For conducting the simulation experiment, 10 number of instances are considered. For each instance, the amount of water leakage is varied. While considering number of pressure data is 250, the amount of water wasted in proposed EBBC-GWO Method is 11lps. Whereas, the existing 1D-CNN-SVM model (Kang et al. 2018) and Multi-Stage Graph Partitioning Approach (Rajeswaran et al. 2018) outflows 15 lps and 13 lps respectively. From that the EBBC-GWO Method reduces the leakage size by 20%

and 12% compared to existing 1D-CNN-SVM model (Kang et al. 2018) and Multi-Stage Graph Partitioning Approach (Rajeswaran et al. 2018) respectively.

**5 Conclusion**

An EBBC-GWO Method is introduced for leakage detection in water distribution system. Enhanced BrownBoost Classifier model taken k-NN classifier as weak classifier and classifies training samples through majority vote of neighbor. After that, Brownboost classifier combines all k-NN classifier output to build the strong classifier. By this way, data are classified as the normal data or abnormal. After classification, glowworm swarm optimization process is carried out where every solution corresponds to abnormal pressure data node in search space. Every glowworm has objective function for addressing the optimization issue. Every glowworm employs probabilistic way to choose neighbor with higher objective function value and moves toward it. Glowworm updates its location to the glowworm in dynamic decision space and optimal one is selected for senor placement to detect the leakage. By this way, water leakage detection accuracy is improved with lesser false positive rate. The performance of EBBC-GWO method is carried out with the parameters such as classification accuracy, false positive rate, classification time, leakage size and water leakage detection accuracy. The results analysis of EBBC-GWO method improves classification accuracy and water leakage detection accuracy with minimum false positive rate and classification time than the state-of-art methods.